# The Psu protein of phage satellite P4 inhibits transcription termination factor ρ by forced hyper-oligomerization

Daniela Gjorgjevikj [1,6], Naveen Kumar [2], Bing Wang [3], Tarek Hilal [1,4], Nelly Said [1], Bernhard Loll [1], Irina Artsimovitch [3], Ranjan Sen[2] & Markus C. Wahl [1,5] ✉

Many bacteriophages modulate host transcription to favor expression of their own genomes. Phage satellite P4 polarity suppression protein, Psu, a building block of the viral capsid, inhibits hexameric transcription termination factor, ρ, by presently unknown mechanisms. Our cryogenic electron microscopy structures of ρ-Psu complexes show that Psu dimers clamp two inactive, open ρ rings and promote their expansion to higher-oligomeric states. ATPase, nucleotide binding and nucleic acid binding studies revealed that Psu hinders ρ ring closure and traps nucleotides in their binding pockets on ρ. Structure-guided mutagenesis in combination with growth, pull-down, and termination assays further delineated the functional ρ-Psu interfaces in vivo. Bioinformatic analyses revealed that Psu is associated with a wide variety of phage defense systems across *Enterobacteriaceae*, suggesting that Psu may regulate expression of anti-phage genes. Our findings show that modulation of the ρ oligomeric state via diverse strategies is a pervasive gene regulatory principle in bacteria.

More than 90% of bacterial species encode ρ, a hexameric, ring-shaped RNA-dependent NTPase[1]. ρ is long known as transcription termination factor[2] that defines the ends of 20–30% of transcription units in *Escherichia coli*[3]. ρ also mediates attenuation in 5'-untranslated regions (UTRs)[4], limits antisense transcription[5,6], and safeguards bacterial genomes by restricting R-loops[7]. A ρ protomer comprises N-terminal and C-terminal domains (NTD and CTD) connected by a flexible hinge region[8,9]. The NTD harbors a primary RNA-binding site (PBS)[8,9]. The CTDs together form a secondary RNA-binding site (SBS) in the central pore of the hexamer and six nucleotide-binding pockets between the subunits[8,9]. The ρ hexamer adopts an NTPase-inactive, washer-like, open-ring conformation until NTPs (preferentially ATP) are engaged and RNA is bound at the SBS, leading to ring closure and RNA

entrapment inside the ring[10]. The SBS and the sixth nucleotide-binding site are fully formed only in the closed ρ ring. After RNA engagement at the PBS and the SBS, ρ can act as an NTP hydrolysis-driven 5'-to-3' RNA-translocase and helicase[8,9].

Extensive research over decades suggested two mechanisms of ρ termination. In a tethered-tracking model, ρ in the open conformation engages a C-rich, secondary-structure-deficient ρ-utilization (*rut*) site on the nascent RNA (a ρ-dependent terminator) via its PBSes[8,9]. Subsequently, downstream RNA can enter the SBS, followed by ring closure and NTPase-driven 5'-to-3' translocation on the transcript, during which the PBSes are thought to remain bound to *rut*[8,9]. When RNAP reaches a pause signal, ρ catches up and, using its strong motor activity, disassembles the elongation complex (EC)[11–13]. In an

---

[1]Laboratory of Structural Biochemistry, Institute of Chemistry and Biochemistry, Freie Universität Berlin, Berlin, Germany. [2]Laboratory of Transcription, Centre for DNA Fingerprinting and Diagnostics, Hyderabad, India. [3]Department of Microbiology and Center for RNA Biology, The Ohio State University, Columbus, OH, USA. [4]Research Center of Electron Microscopy and Core Facility BioSupraMol, Institute of Chemistry and Biochemistry, Freie Universität Berlin, Berlin, Germany. [5]Helmholtz-Zentrum Berlin für Materialien und Energie, Macromolecular Crystallography, Berlin, Germany. [6]Present address: Department of Medicine, Molecular Immunity Unit, MRC Laboratory of Molecular Biology, University of Cambridge, Cambridge, UK. ✉e-mail: markus.wahl@fu-berlin.de

alternative, allosteric model, the NTDs of an open ρ ring engage ECs via direct contacts to RNAP and bound general transcription factors (TFs) NusA and NusG[14–16] and ρ traffics with the EC[17]. Once the EC pauses, ρ can stepwise inactivate RNAP without resorting to its motor activity[14–16]. Recent single-molecule spectroscopic studies have suggested that, at least in vitro, both mechanisms may be at work to a different extent depending on the transcription unit or termination scenario[18,19].

A major function of ρ is the silencing of horizontally acquired genes and invading foreign DNA[20]. Thus, phages that utilize the bacterial transcription machinery must counteract ρ activity to efficiently express their genomes. While lambdoid phages implement protein- or RNA-based anti-termination mechanisms to render the host RNA polymerase (RNAP) termination-resistant[21–27], the enterobacterial phage satellite, P4, has acquired a unique anti-termination system that acts directly upon ρ. P4 polarity suppression (Psu) protein is a building block of its capsid[28], but can also inhibit ρ[29,30]. Phage satellites are virus-like entities that require infection by another (helper) phage to reproduce. The majority of phage satellites, including P4, hijack structural proteins of helper phages to assemble viral capsids of reduced size. The small capsids can package the satellite genomes but exclude the larger helper genomes, facilitating the selective mobilization of the phage satellite for infection of new host cells. To this end, P4 protein Sid diverts structural proteins of the P2 helper phage[31,32]. P4 virion production benefits from Psu-mediated suppression of ρ-dependent polarity[33] in the P2 morphogenic gene operons[34], which gives rise to an increased ratio of phage tail to capsid morphogenesis components[35]. Linking changes in expression of structural genes to capsid remodeling may be a common strategy employed by phage satellites[36,37].

Previous studies reported that Psu decreases ρ ATP binding and hydrolysis, and hypothesized that Psu poses an obstacle to RNA translocation[38,39]. Mutational and cross-linking analysis mapped interaction surfaces on Psu and ρ[38,40], based on which it was proposed that Psu covers the central pore of a closed ρ ring[38]. However, the precise mode of Psu-dependent ρ inhibition has so far remained elusive.

Here, we present cryogenic electron microscopy (cryoEM)/single-particle analysis (SPA)-based structures of ρ-Psu complexes, defining the precise ρ-Psu interfaces and revealing a unique inhibitory mechanism. Multiple Psu dimers staple two open ρ complexes together and promote the formation of higher-order ρ oligomers. Psu subunits bind between two neighboring ρ CTDs across the nucleotide-binding sites. Our systematic ATPase, nucleotide-binding and nucleic-acid-binding studies are fully consistent with Psu blocking the ρ nucleotide-binding sites and stabilizing ρ in an open conformation, while leaving ρ PBSes unobstructed. Furthermore, we delineate key residues on Psu and ρ required for ρ inhibition in vitro and in vivo. We also find Psu proteins encoded in the neighborhood of known and putative phage defense genes in many bacterial genomes. Based on our results, we propose that Psu inhibits ρ-dependent transcription termination via an unconventional molecular mechanism of forced hyper-oligomerization, which may well be employed by diverse bacteria to regulate the expression of phage defense systems.

## Results

### Psu dimers bridge and promote the expansion of two open ρ hexamers

Mixtures of ρ and Psu started to elute earlier during analytical size-exclusion chromatography (SEC) than ρ alone, and the start of the elution shifted to yet smaller volumes in the presence of ATP (Supplementary Fig. 1a) or the analogs ATPγS or ADP-BeF$_3$. These observations suggest that ρ and Psu form dynamic complexes that are reinforced by ATP or ATP analogs but fail to migrate as stable assemblies in SEC. To further test this notion, we conducted dynamic light scattering (DLS) analyses with ρ, Psu, and ρ/Psu mixtures in the

absence or presence of ATP, ATPγS or ADP-BeF$_3$ (Supplementary Fig. 1b). All samples showed low polydispersity indices (Supplementary Fig. 1b), indicating that each sample comprised a narrow distribution of particle populations in solution. ρ exhibited a hydrodynamic radius ($r_H$) of 5.50 nm, in good agreement with the theoretical values of 5.90 and 5.97 nm for closed (PDB ID: 5JJI; RNA ligand omitted) and open (PDB ID: 6WA8) ρ hexamers calculated with the HullRad algorithm[41]. When ρ was incubated with Psu in the absence or presence of ATP, ATPγS or ADP-BeF$_3$, $r_H$ values increased to 7.92–8.66 nm, much larger than expected for the previously modeled ρ-Psu complex (calculated $r_H$ = 6.33 nm). Taken together, these observations suggest that Psu induces the formation of dynamic, high-molecular mass ρ-Psu complexes that are stabilized by ATP or analogs.

To elucidate the structural basis of nucleotide-supported ρ-Psu complex formation, we subjected ρ/ATP/Psu and ρ/ATPγS/Psu mixtures to cryoEM/SPA (Supplementary Table 1; Supplementary Figs. 2–6). In both cases, multi-particle 3D refinement yielded a series of cryoEM reconstructions representing large ρ-ATP/ATPγS-Psu complexes (Supplementary Figs. 3 and 5). Irrespective of the bound nucleotide, all assemblies contain two open ρ rings, denominated ρ(A) and ρ(B), that are laterally bridged by multiple Psu dimers, with the ρ CTDs facing inwards and the ρ NTDs facing outwards (Figs. 1 and 2). The assemblies differ by the number of Psu dimers that bridge the ρ rings and by the number of subunits in the Psu-bridged ρ rings.

In assemblies comprising two hexameric ρ rings, the different numbers of interconnecting Psu dimers are due to the different rotational orientations of the two ρ rings. We designate individual subunits of ρ(A)/ρ(B) by increasing Arabic numerals from the inner-most to the outer-most subunits. In complex I, two hexameric ρ rings are bridged by four Psu dimers that connect ρ(A) subunit 6 with ρ(B) subunit 3, ρ(A) subunit 5 with ρ(B) subunit 4, etc. (Fig. 1a). In complex II, two hexameric ρ rings are bridged by five Psu dimers that connect ρ(A) subunit 6 with ρ(B) subunit 2, ρ(A) subunit 5 with ρ(B) subunit 3, etc. (Fig. 1b). In complex III, two hexameric ρ rings are bridged by six Psu dimers, that connect ρ(A) subunit 6 with ρ(B) subunit 1, ρ(A) subunit 5 with ρ(B) subunit 2, etc. (Figs. 1c and 2a). The different rotational orientations of the Psu-bridged ρ rings in complexes I, II, and III lead to different spacings between the innermost ρ subunits, which directly interact in complex I but are separated by a gap of ~30 Å and ~50 Å in complexes II and III, respectively. Other observed assemblies are partially dissociated or further expanded versions of these basic complexes. Complex II$^{expanded}$ is a complex II constellation, in which both ρ rings are augmented to the octamer level by incorporation of two additional peripheral ρ subunits, providing binding sites for two additional Psu dimers (Fig. 1d). A yet larger assembly, complex III$^{expanded}$, was observed in the ρ-ATPγS-Psu sample (Fig. 1e); the complex III arrangement leaves sufficient space in the center for incorporation of an additional subunit at each ρ ring (subunits 0 in Figs. 1e and 2b), and the rings are further expanded at the peripheries by two ρ subunits (subunits 7 and 8 in Figs. 1e and 2b); again, the additional ρ subunits generate binding sites for additional Psu dimers (Fig. 2b, c). The central Psu dimers of complex III$^{expanded}$ are spaced close enough to reciprocally interact via their C-terminal α7 helices (Fig. 2b). Finally, complex III$^{fragmented}$ can be considered a partially assembled complex III, with only three bridging Psu dimers (Fig. 1f). Irrespectivley, the observed higher-order ρ-Psu complexes are consistent with the large hydrodynamic radii observed in the DLS analyses (Supplementary Fig. 1b).

We further refined complex II (ATP; global resolution 3.8 Å), complex II$^{expanded}$ (ATP; global resolution 3.6 Å), complex III (ATPγS; global resolution 3.65 Å), and complex III$^{expanded}$ (ATPγS; global resolution 4.25 Å). The most peripheral ρ subunits of the ρ(A)/ρ(B) rings in complexes II$^{expanded}$ and III$^{expanded}$ are not stabilized by contacts to a Psu dimer, as no appropriately-spaced, unoccupied ρ subunit remains in the opposite ρ ring. While clearly defined in low-pass-filtered maps (6 Å

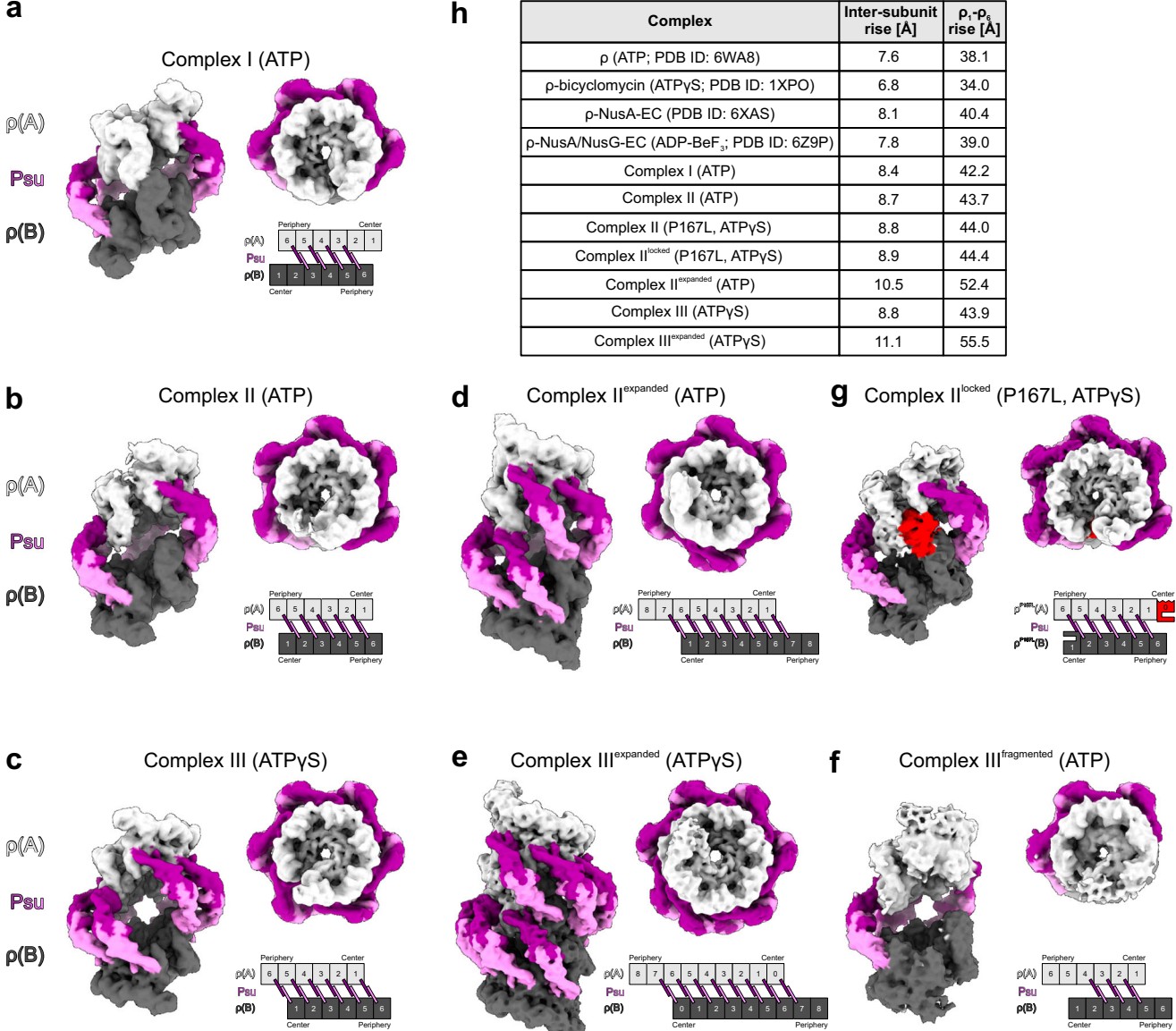

**Fig. 1 | Nucleotide-stabilized ρ-Psu complexes. a–g** 6 Å resolution, low-pass-filtered cryoEM reconstructions of different classes of ρ-Psu complexes observed in the present study. Each panel shows a reconstruction in side view (left) and top view (right) and the corresponding ρ-Psu interaction scheme. ρ subunits at the centers or the peripheries of the complexes are indicated in the schemes. Bound nucleotides and, if applicable, ρ residue changes are indicated in parentheses. Complexes I, II and III contain hexameric ρ rings, ρ(A) and ρ(B), in different relative orientations bridged by an increasing number of Psu dimers. The complex II and III constellations can be further expanded by incorporation of additional ρ subunits and Psu dimers (complex II^expanded, complex III^expanded, complex II^locked). ρ(A), light gray; ρ(B) dark gray; interlocking ρP167L subunit in complex II^locked, red; Psu dimers, purple and violet. **h** ρ helical parameters for the indicated complexes. Parameters listed for ρ^P167L-ATPγS-Psu complex II^locked refer to the $\rho_1$-$\rho_6$ subunits of ρ(A), which retain the standard conformation.

| Complex | Inter-subunit rise [Å] | $\rho_1$-$\rho_6$ rise [Å] |
|---|---|---|
| ρ (ATP; PDB ID: 6WA8) | 7.6 | 38.1 |
| ρ-bicyclomycin (ATPγS; PDB ID: 1XPO) | 6.8 | 34.0 |
| ρ-NusA-EC (PDB ID: 6XAS) | 8.1 | 40.4 |
| ρ-NusA/NusG-EC (ADP-BeF$_3$; PDB ID: 6Z9P) | 7.8 | 39.0 |
| Complex I (ATP) | 8.4 | 42.2 |
| Complex II (ATP) | 8.7 | 43.7 |
| Complex II (P167L, ATPγS) | 8.8 | 44.0 |
| Complex II^locked (P167L, ATPγS) | 8.9 | 44.4 |
| Complex II^expanded (ATP) | 10.5 | 52.4 |
| Complex III (ATPγS) | 8.8 | 43.9 |
| Complex III^expanded (ATPγS) | 11.1 | 55.5 |

resolution; Figs. 1d, e and 2b), density for these peripheral subunits is weak and fragmented in the cryoEM reconstructions refined to full resolution, suggesting that they are more flexibly connected or bound sub-stoichiometrically. We nevertheless retained these peripheral subunits in the final models.

Isolated open ρ hexamers bound to ATP (PDB ID: 6WA8)[42] exhibit an average inter-subunit rise of 7.6 Å, yielding an overall $\rho_1$-$\rho_6$ rise of 38.1 Å (Fig. 1h), which are too small to allow oligomerization beyond six ρ protomers. The ρ hexamers in complexes I, II and III exhibit slightly increased inter-subunit and $\rho_1$-$\rho_6$ rise values (8.4–8.8 Å and 42.2–43.9 Å, respectively; Fig. 1h). The incorporation of additional ρ subunits supported by additional, bridging Psu dimers in complexes II^expanded and III^expanded is facilitated by a further increase in the inter-subunit and $\rho_1$-$\rho_6$ rise values (10.5/11.1 Å and 52.4/55.5 Å, respectively;

thus, the helical pitch in complexes II^expanded and III^expanded is large enough to allow expansion by additional ρ protomers, as they can stack on preceding subunits in the stretched ρ rings.

## Structural basis for increased stability of ρ^P167L-Psu complexes

A ρ variant harboring a P167L substitution has been observed to bind Psu more tightly than wt ρ[38,43]. Consistently, we observed a more pronounced shift in the SEC elution volume for ρ^P167L-Psu mixtures, even in the absence of nucleotides, as compared to wt ρ (Supplementary Fig. 1c). DLS analysis suggested that ρ^P167L forms similar-size complexes with Psu as wt ρ (Supplementary Fig. 1b). However, in the ρ-ATP/ATPγS-Psu structures, P167 of ρ does not engage in direct contacts to Psu that could explain the higher complex stability. We, therefore, also determined cryoEM structures of ρ^P167L-ATPγS-Psu

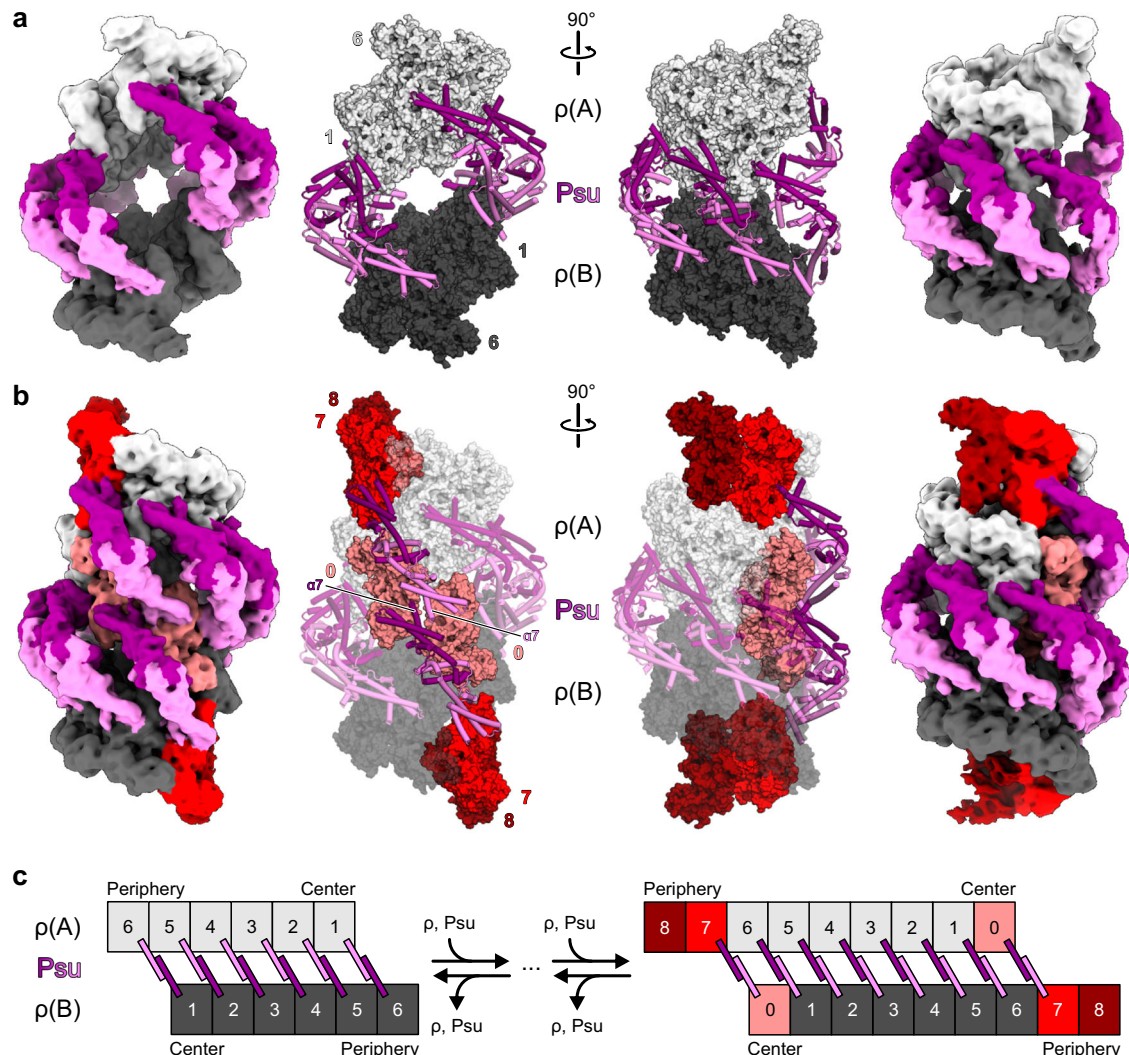

**Fig. 2 | Structures of ρ-ATPγS-Psu complexes. a** Orthogonal views of the cryoEM reconstruction (left and right; 6 Å resolution, low-pass-filtered) and model (center) of ρ-ATPγS-Psu complex III. Six Psu dimers bridge two open ρ hexamers, ρ(A) and ρ(B). ρ subunits are designated by increasing Arabic numerals from the centers to the peripheries of the complex. Coloring as in Fig. 1. **b** Orthogonal views of the cryoEM reconstruction (left and right; 6 Å resolution, low-pass-filtered) and model (center) of ρ-ATPγS-Psu complex III^expanded. Eight Psu dimers (purple and violet) bridge two open ρ nonamers, ρ(A) and ρ(B). Portions of ρ(A) and ρ(B) equivalent to ρ-ATPγS-Psu complex III are shown in the same colors as in (**a**). In the model, the portion of complex III^expanded that is equivalent to complex III is shown in semi-transparent cartoon/surface representation, additional ρ subunits and Psu dimers in complex III^expanded are shown in solid surface and cartoon representation. ρ-ATPγS-Psu complex III^expanded can be envisaged to emerge from complex III by the addition of ρ subunits at the center and periphery of complex III (shades of red) and additional bridging Psu dimers. **c** Schemes illustrating ρ-Psu interaction patterns in ρ-ATPγS-Psu complexes III and III^expanded and how ρ-ATPγS-Psu complexes III^expanded could emerge from complexes III.

complexes (Supplementary Table 1; Supplementary Figs. 6–9). The results showed that ρ^P167L associates with Psu in a similar manner as wt ρ, with a preference for the complex II constellation (Supplementary Fig. 8). In this arrangement, a single additional ρ^P167L subunit can join one ring at the center of the complex (complex II^locked; Fig. 1g; Supplementary Fig. 9a–c). Incorporation of this additional ρ^P167L subunit leads to an interlocking of the two ρ^P167L rings, due to refolding and domain swapping of the central ρ^P167L subunits, most likely explaining the increased ρ^P167L-Psu affinity (Supplementary Discussion; Supplementary Fig. 9d). Such refolding and domain-swapping in ρ^P167L only occurs in complex with Psu, as it is not observed in a cryoEM structure of isolated ρ^P167L (Supplementary Table 1; Supplementary Figs. 6, 10 and 11).

**Psu binds across ρ nucleotide-binding pockets**

ATP or ATPγS are bound similarly between all neighboring ρ CTDs in the ρ-ATP/ATPγS-Psu or ρ^P167L-ATPγS-Psu assemblies (Fig. 3a). The nucleotides are bound as previously observed for open ρ complexes[42] (Fig. 3b). Thus, the adenine base is sandwiched between F355 and M186, K181 contacts the γ-phosphate and T185 together with the β/γ-phosphates coordinates the Mg^2+ ion (Fig. 3b). The γ-phosphate is additionally bound by R212 (arginine valve), and by R366 (arginine finger) of the adjacent ρ CTD (Fig. 3b). Compared to the hydrolytic situation, the catalytic glutamate (E211) and the Walker B aspartate (D265) are displaced from the ATP phosphates and there is no evidence for a bound attacking water molecule.

Except at the inner-most ρ(A/B) subunits of complexes II^expanded and III^expanded, each Psu molecule binds in a similar fashion between two neighboring ρ CTDs in all complexes observed (root-mean-square deviations of 0.6–2.0 Å for ~8100 atom pairs upon global, pairwise superposition of ρ-ρ-Psu sub-complexes; Fig. 3a). Psu interacts predominantly with one ρ CTD via the tip of its long coiled-coil (helices α1 and α2), the N-terminal part of helix α2 and the C-terminal helix α7. Psu helix α7 (residues 170–186) is placed in a cleft between the two

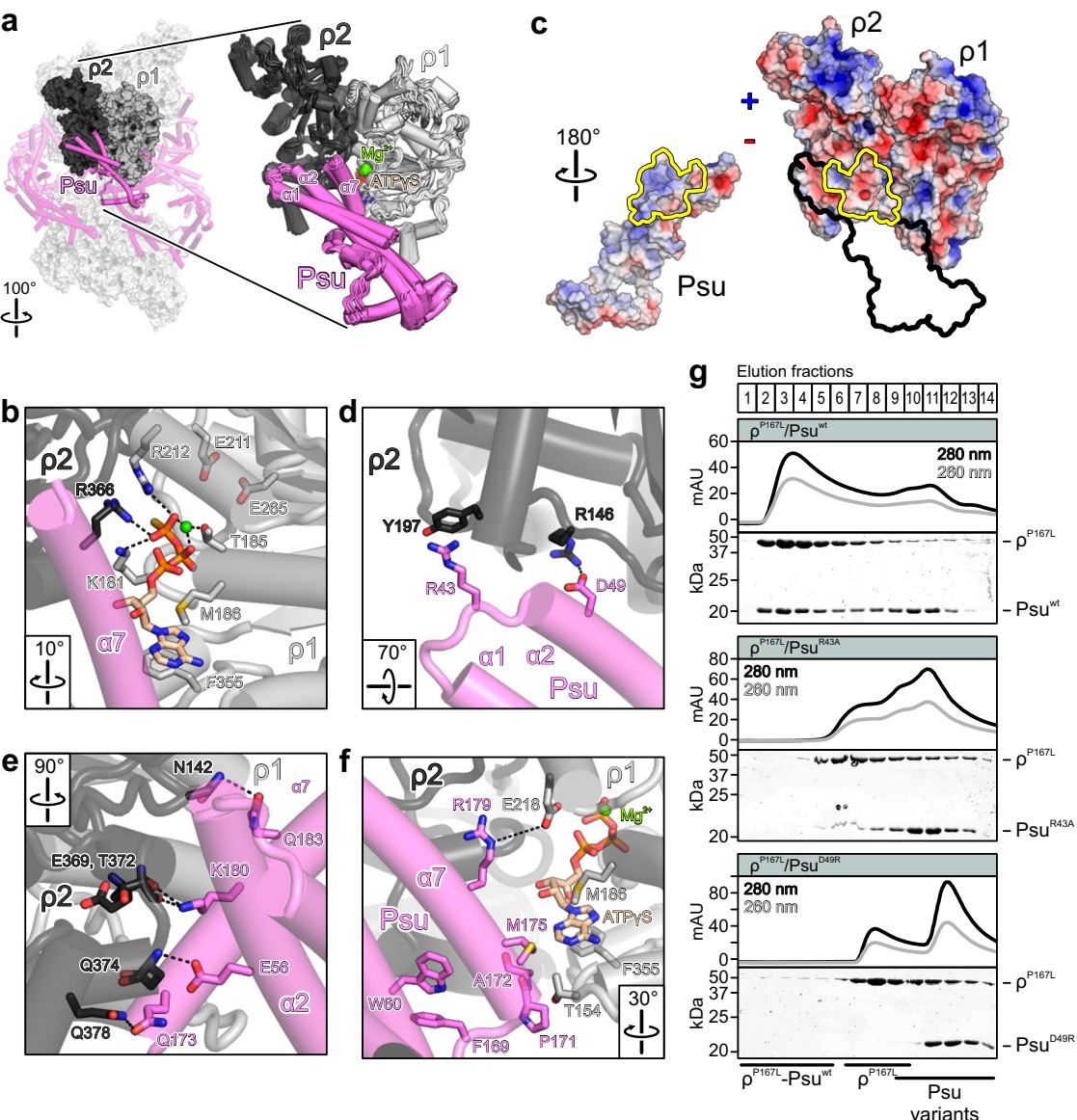

**Fig. 3 | ρ-Psu interaction details. a** Left, ρ-ATPγS-Psu complex III with a building block comprising two neighboring ρ subunits (ρ1 and ρ2) and an interacting Psu molecule highlighted in a solid representation. Right, overlay of all corresponding building blocks of the ρ-ATPγS-Psu complexes III and III[expanded]. ρ-interacting helices of Psu are labeled. Bound ATPγS molecules and Mg²⁺ ions are shown as sticks and spheres, respectively, colored by atom type. In this and the following panels: ATPγS carbon, beige; nitrogen, blue; oxygen, red; sulfur, yellow; phosphorus, orange; magnesium, green. Rotation symbol, view relative to Fig. 2a, left. **b** Details of ATPγS binding at the building blocks shown in (**a**). Relevant protein residues are shown as sticks and colored by atom type. In this and the following panels: carbon, as the respective protein subunit; black dashed lines, hydrogen bonds or salt bridges; rotation symbols, view relative to (**a**). **c** Book-view on the interacting surfaces (yellow outlines) of Psu (left) and of two neighboring ρ subunits (right), with the electrostatic potential (blue, positive; red, negative) mapped onto the protein surfaces. Black outline, border of the bound Psu molecule. **d, e, f** Details of the ρ-Psu interaction within the building blocks shown in (**a**). **g** Elution profiles (top) and SDS-PAGE analyses (bottom) of analytical SEC elution fractions monitoring the interaction of ρ[P167L] with Psu variants. Elution fractions are indicated at the top. The same elution fractions were analyzed for each run and were aligned below each other. Proteins and protein mixtures analyzed are indicated above each gel. Molecular mass markers are indicated on the left. Protein bands are identified on the right. Fractions containing isolated proteins or complexes are identified at the bottom. Experiments were repeated independently at least two times with similar results. Source data are provided as a Source Data file.

neighboring ρ CTDs, covering the corresponding nucleotide-binding pocket (Fig. 3a). The interaction is supported by shape and electrostatic complementarity (Fig. 3c).

Psu R43 at the tip of the coiled-coil engages in cation-π interactions with Y197 of the main interacting ρ subunit (Fig. 3d). Psu D49 at the N-terminus of helix α2 (residues 45-91) forms an electrostatic interaction with ρ R146 (Fig. 3d). Psu E56 (helix α2) hydrogen bonds with the ρ Q374 backbone amide (Fig. 3e). Psu Q173, K180 and Q183 (helix α7) form hydrogen bonds with the ρ Q378 side chain, the backbone carbonyls of ρ E369/T372, and the side chain of ρ N142,

respectively (Fig. 3e). Furthermore, there is a weak electrostatic interaction between Psu R179 (helix α7) and E218 of the adjacent ρ subunit (Fig. 3f). Finally, Psu P171, A172 and M175 (helix α7) engage in contacts with the T154-loop of the neighboring ρ subunit and cover the Watson-Crick side of the ATPγS base bound at that subunit, stabilizing it between ρ M186 and F355 (Fig. 3f). In the largest ρ-ATPγS-Psu complex (complex III[expanded]), additional contacts ensue between the dimerization region of the central Psu dimers and the NTDs of the central ρ subunits (Fig. 2b). Phage satellite P4 host range is restricted to *Enterobacteriacea* and virtually all *psu* genes are found in

*Enterobacteriaceae* (Supplementary Data 1) which likely is one of the reasons why Psu-binding residues in ρ lack conservation across all bacteria (Supplementary Fig. 12a). The conservation pattern of ρ-interacting residues in Psu supports the functional importance of helix α7-ρ contacts revealed by the structure: Psu residues P171, A172 and Q173 are highly conserved, and only K or R are found at position 180 (Supplementary Fig. 12b).

## ρ-contacting residues of Psu are required for complex formation and inhibition of ρ ATPase activity

Effects of previously investigated Psu and ρ variants[28,33,38,39,44] are fully reconciled by our structures (Supplementary Table 2). To further test the importance of specific contacts observed in our structures, we exchanged Psu residues R43 and D49 (that contact ρ Y197 and R146, respectively; Fig. 3d), to alanine and arginine, respectively, and tested the binding of these variants to $\rho^{P167L}$ that forms stable complexes with Psu in SEC independent of added nucleotides. $Psu^{R43A}$ showed reduced binding to $\rho^{P167L}$ in analytical SEC, as indicated by a reduced shift in elution volume compared to $\rho^{P167L}$-$Psu^{wt}$, while binding of $Psu^{D49R}$ was completely abolished (Fig. 3g).

We next investigated Psu effects on ρ ATPase activity[39] by measuring ATP hydrolysis by $\rho^{wt}$ and $\rho^{P167L}$, stimulated by RNA containing the *rut* site of the strong $\lambda t_{R1}$ terminator. Reaction mixtures contained a fixed amount of ρ variants, increasing amounts of Psu, and a large (~10,000-fold) excess of ATP. Reactions were started by the addition of ATP, samples were taken at various time points, and nucleotides were separated by thin-layer chromatography (TLC; Source Data) and quantified. The time traces of hydrolyzed ATP could be fitted to a single exponential equation (Fig. 4a). The final percentage of hydrolyzed ATP (reaction amplitude) was comparable for $\rho^{wt}$ and $\rho^{P167L}$, but the ATP turnover rate of $\rho^{P167L}$ was approximately doubled compared to $\rho^{wt}$ (Fig. 4a). For both ρ variants, addition of increasing amounts of wt Psu (Psu dimer:ρ hexamer 10:1 and 50:1) led to the stepwise decrease of the ρ ATPase rate (1.3-fold and 1.6-fold, respectively, for $\rho^{wt}$; 1.5-fold and 4-fold, respectively, for $\rho^{P167L}$) and amplitude (by 20% and 70%, respectively, for $\rho^{wt}$; by 45% and 70%, respectively, for $\rho^{P167L}$; Fig. 4a). Psu inhibited RNA-stimulated ATPase activities of $\rho^{wt}$ and $\rho^{P167L}$ with comparable $IC_{50}$ values (0.9 μM and 0.8 μM, respectively; Fig. 4b). In contrast, neither $Psu^{R43A}$ nor $Psu^{D49R}$ significantly affected the ATPase activity of wt ρ, and these variants were partially ($Psu^{R43A}$; $IC_{50}$ = 12.9 μM) or fully ($Psu^{D49R}$) defective in inhibiting the $\rho^{P167L}$ ATPase (Fig. 4b). Together, the above results confirm the importance of specific Psu residues in establishing contacts to ρ that inhibit the RNA-stimulated ATPase activity of ρ, fully consistent with our structures.

## Psu effectively inhibits nucleotide trafficking on ρ

As Psu helix α7 covers nucleotide-binding sites of ρ in our structures (Fig. 3a, b), Psu may hinder nucleotide binding to and release from ρ. To examine this possibility, we monitored mant-ATPγS binding via Förster resonance energy transfer (FRET) from ρ W381 in the vicinity of the nucleotide-binding pockets in a stopped-flow device. To this end, we quickly mixed constant amounts of ρ variants or pre-assembled $\rho^{P167L}$-Psu complex with increasing concentrations of mant-ATPγS at 18 °C. For each mant-ATPγS concentration, the apparent rate constants ($k_{app}$) of binding were determined by fitting the time traces of fluorescence changes to a single exponential equation (Fig. 5a). The $k_{app}$ values were plotted against the nucleotide concentrations and fitted to a linear regression function to estimate the association rate constant ($k_{on}$; slope) and the dissociation rate constant ($k_{off}$, y-intercept; Fig. 5b). Dissociation constants ($K_d$) were calculated as the ratios of the $k_{off}$ and $k_{on}$ values (Fig. 5b).

In the absence of Psu, $\rho^{wt}$ and $\rho^{P167L}$ exhibited similar $K_d$'s for mant-ATPγS (9.3 μM and 14.0 μM, respectively; Fig. 5b). The slightly lower mant-ATPγS affinity of $\rho^{P167L}$ is due to a two-fold higher dissociation rate (Fig. 5b), which likely arises because the residue substitution

renders the nucleotide-binding site more dynamic, as indicated by the lower thermal stability of $\rho^{P167L}$ (Supplementary Fig. 9e). The increased nucleotide dissociation constant may enable faster ATP turnover, consistent with the observed elevated ATPase rate of $\rho^{P167L}$ compared to $\rho^{wt}$ (Fig. 4a).

The time-dependent change in the FRET signal observed with $\rho^{P167L}$-Psu was almost five-fold stronger than for $\rho^{P167L}$ alone (Fig. 5a). We attribute this effect to bound Psu, which contributes additional FRET donors (W20, W60) in the vicinity of the $\rho^{P167L}$ nucleotide-binding pockets, close to or below the tryptophan-mant Förster radius of ~25 Å[45], thus enhancing the energy transfer to the mant-moiety. In cryoEM reconstructions of $\rho^{wt/P167L}$-ATPγS-Psu complexes, we observed density features suggesting covalent attachment of ATPγS to C117 of Psu upon prolonged incubation. In principle, a similar covalent attachment of mant-ATPγS could contribute to the observed increase in FRET signal, as Psu residues W20, W60, and W133 reside within a distance of ~25 Å or less of C117. However, mixing Psu and mant-ATPγS did not lead to a significant FRET signal in the timeframe of the experiment.

The $\rho^{P167L}$-Psu complex exhibited a ~seven-fold higher mant-ATPγS affinity compared to $\rho^{P167L}$ alone ($K_d$ = 2.1 μM; Fig. 5b). The mant-ATPγS association and dissociation rates of the complex were more than five-fold and ~40-fold lower, respectively, compared to $\rho^{P167L}$ alone (Fig. 5b). The drastic decrease of the mant-ATPγS dissociation rate in the presence of Psu suggests that, once nucleotide is bound, Psu effectively hinders its release.

Under the conditions used above, we did not observe Psu-dependent changes in the FRET signal when quickly mixing mant-ATPγS with $\rho^{wt}$/Psu. We, therefore, increased the reaction temperature to 37 °C to accelerate binding events and acquired longer time traces to monitor possible slower interactions. Under these conditions, we observed an expected temperature-dependent increase of the $k_{app}$ for mant-ATPγS binding (about four-fold compared to 18 °C; Fig. 5c, first panel). In the presence of Psu, two phases could be clearly distinguished and the time-dependent FRET signal could be fitted to a double-exponential function (Fig. 5c, second panel). The apparent rate of the fast phase ($k_{app1}$) was equivalent to that of mant-ATPγS binding to ρ alone, indicating mant-ATPγS binding to unmodified ρ (Fig. 5c, first and second panels). As the slow phase ($k_{app2}$) was associated with a further increase in the FRET signal, similar to what was observed with $\rho^{P167L}$ in the presence of Psu, it likely monitored the subsequent binding of Psu to the ρ-mant-ATPγS complex. To test this notion, we pre-assembled ρ-mant-ATPγS complex and quickly mixed it with Psu in the stopped-flow instrument. The resulting time-dependent FRET signal could be fitted with a single exponential equation, which indicated an apparent rate of Psu binding in perfect correspondence to the slow phase ($k_{app2}$) of the ρ/Psu vs. mant-ATPγS measurement (Fig. 5c, second and third panels). Thus, the $k_{app}$ values derived from the individual ρ to mant-ATPγS and ρ-mant-ATPγS to Psu binding assays are equivalent to the fast rate constant, $k_{app1}$, and the slow rate constant, $k_{app2}$, respectively, derived from the double exponential fit of the ρ/Psu to mant-ATPγS binding assay. Finally, we mixed an excess of non-labeled ATPγS with pre-formed ρ-mant-ATPγS or ρ-mant-ATPγS-Psu complexes to quantify the mant-ATPγS dissociation rates. The presence of Psu decreased the apparent dissociation rate constant by more than 20-fold (Fig. 5d).

Collectively, these results confirm the higher affinity of $\rho^{P167L}$ to Psu compared to wt ρ; corroborate that Psu binding to wt ρ is supported by prior nucleotide binding to ρ; and support the notion that Psu helix α7 obstructs access to, and exit from, the ρ nucleotide-binding pockets. We note that the observed higher affinity of the $\rho^{P167L}$-Psu complex for mant-ATPγS compared to isolated $\rho^{P167L}$ appears to be in conflict with a previously observed Psu-mediated reduction in the ρ-ATP crosslinking efficiency[39]. However, affinities for mant-ATPγS reported here are based on the quantification of both association and dissociation rates, while results from a crosslinking assay will be

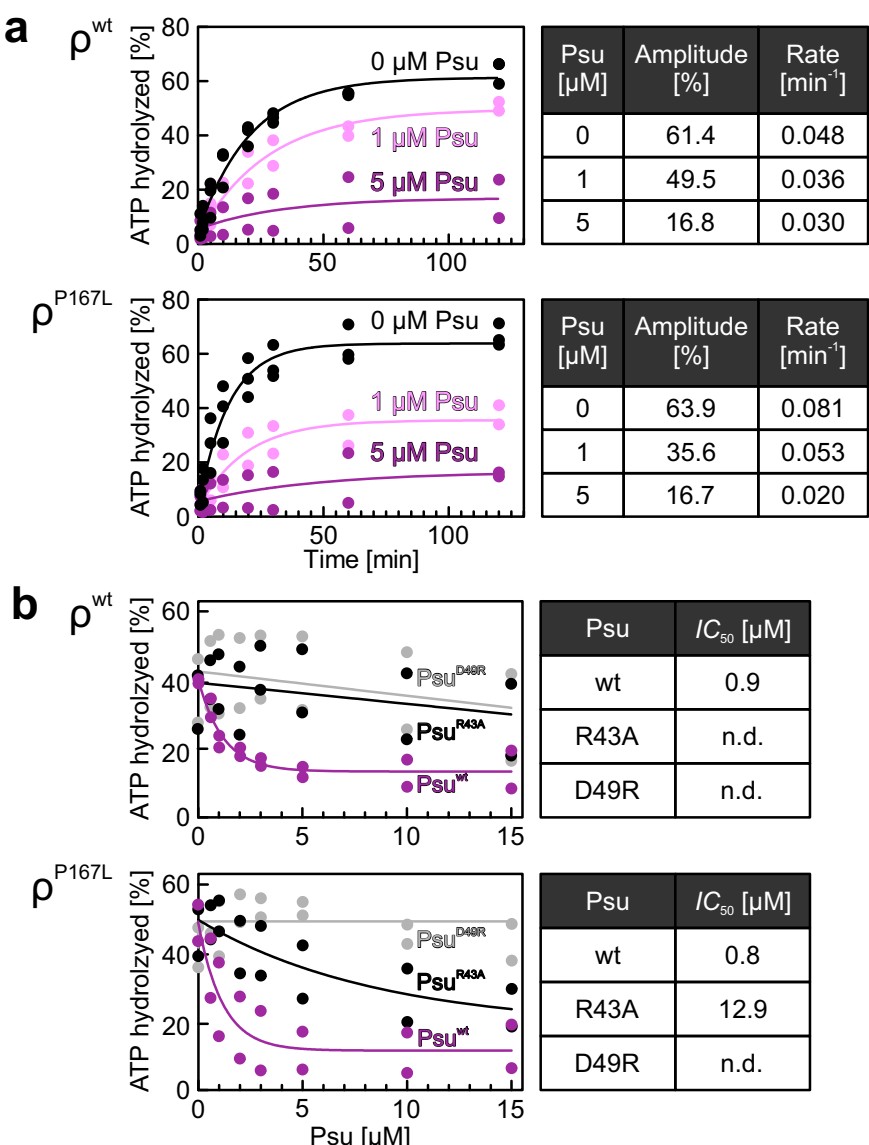

**Fig. 4 | Effect of Psu on ρ ATPase activity. a** Time traces monitoring wt ρ (top) and ρ^P167L (bottom) RNA-stimulated ATPase activities in the absence of Psu or in the presence of 1 or 5 μM of Psu, recorded via TLC. Data were recorded as biological replicates, (ρ^wt or ρ^P167L alone, $n = 3$; ρ^wt or ρ^P167L plus Psu, $n = 2$). Data were fitted to a single-exponential equation; $A = A_0 + (A_{max}-A_0)*(1-\exp(-k*t))$; $A$, fraction of ATP hydrolyzed at time $t$; $A_O$, fraction of ATP hydrolyzed at time zero; $A_{max}$, fraction of ATP hydrolyzed at infinite time (amplitude); $k$, rate constant. Quantified amplitudes and rate constants are listed on the right. Source data are provided as a Source Data file. **b** Inhibition of wt ρ (top) and ρ^P167L (bottom) RNA-stimulated ATPase activities, at 1 h reaction time, by increasing concentrations of Psu^wt, Psu^R43A or Psu^D49R. Data represent biological replicates, $n = 2$. Data were fitted to an [inhibitor] vs. response function, $A = A_{min} + (A_{max} - A_{min}) / (1 + ([inhibitor]/IC_{50}))$; $A$, fraction of ATP hydrolyzed at a given Psu (inhibitor) concentration; $A_{min}$ and $A_{max}$, fitted minimum and maximum fractions of ATP hydrolyzed. Quantified $IC_{50}$ values are listed on the right; n.d., not determined. Source data are provided as a Source Data file.

dominated by differences in the association rates, for which we also observed a Psu-dependent reduction in our assays. Furthermore, due to the bulky mant moiety, which may establish additional contacts to ρ and to which nucleotide-binding sites may be less accessible compared to unmodified ATP, our results most likely do not reflect authentic ATP binding kinetics and thermodynamics. However, they clearly document a differential accessibility of the ρ nucleotide-binding pockets in the absence and presence of Psu. The reduced nucleotide binding and release in the presence of Psu likely contributes to the observed Psu-mediated inhibition of the ρ ATPase.

**Psu does not alter PBS accessibility but inhibits SBS binding of ρ^P167L**

Stable RNA binding at the ρ SBS and RNA-stimulated ATPase activity of ρ require ring closure[10]. In our ρ-Psu complex structures, ρ invariably adopts an open ring conformation, and we showed that Psu inhibits the ρ ATPase (Fig. 4), suggesting that Psu may also interfere with RNA-SBS interactions. In contrast, ρ NTDs are invariably turned outwards in the structures, in principle granting RNAs access to the ρ PBSes. In agreement with the latter notion, a previous study found that Psu does not affect ρ binding to the PBS ligand poly(dC)[39].

To test the potential effects of Psu on ρ-nucleic acid interactions in more detail, we used fluorescence anisotropy assays[46]. ρ PBSes bind pyrimidine-rich DNA or RNA while the ρ SBS exclusively engages RNA[47]. Thus, for probing PBS binding, we used a fluorescein-labeled DNA oligomer, dC15-FAM. For testing RNA binding to the SBS, we used a fluorescein-labeled RNA oligomer, rU12-FAM, after saturating the PBSes with unlabeled dC15. ADP-BeF3 was included in the assays to stabilize ρ-Psu interactions. PBS and SBS affinities to the respective nucleic acid probes were very similar for ρ^wt and ρ^P167L (Fig. 6a–d).

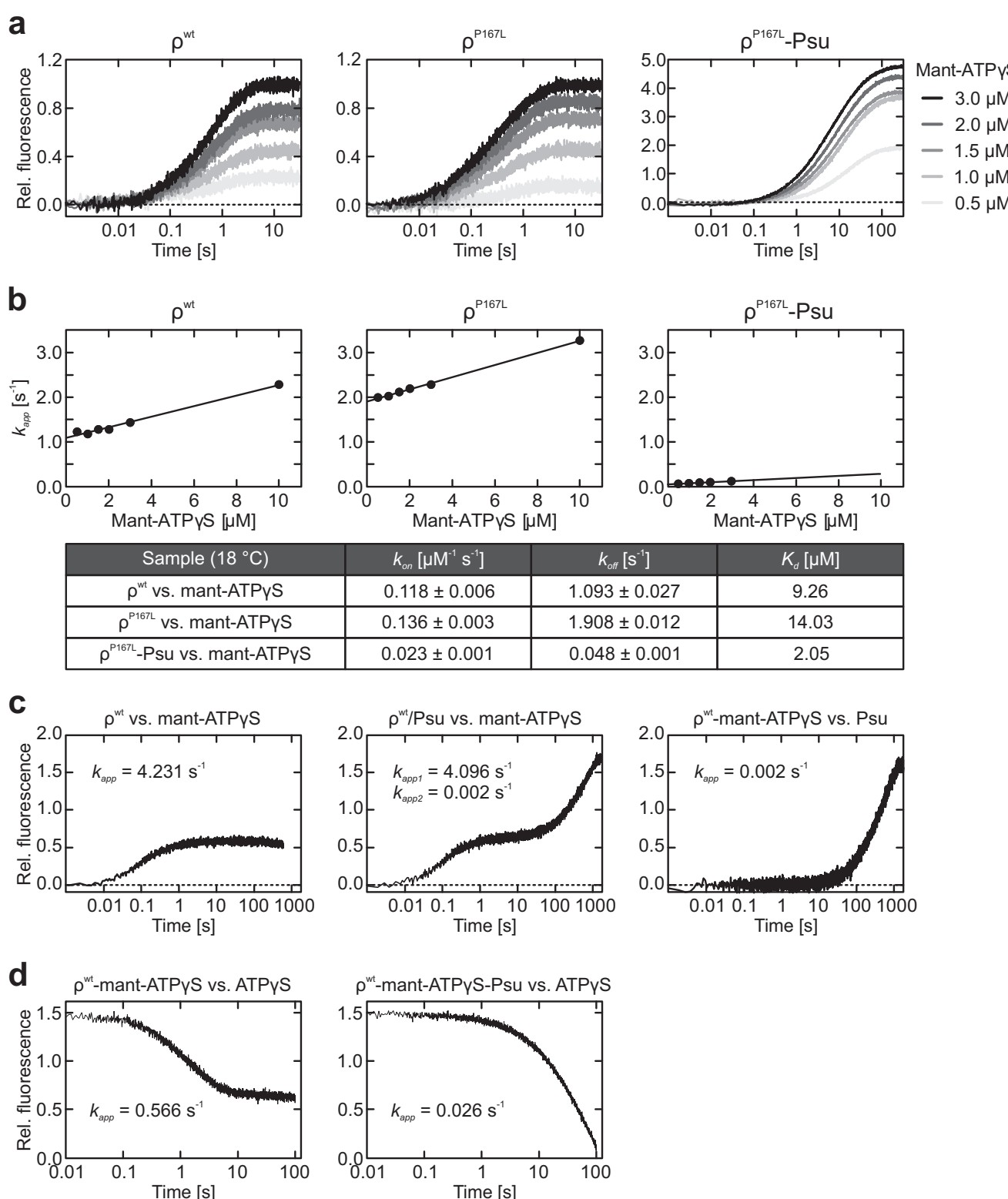

| Sample (18 °C) | $k_{on}$ [$\mu M^{-1} s^{-1}$] | $k_{off}$ [$s^{-1}$] | $K_d$ [$\mu M$] |
|---|---|---|---|
| $\rho^{wt}$ vs. mant-ATPγS | 0.118 ± 0.006 | 1.093 ± 0.027 | 9.26 |
| $\rho^{P167L}$ vs. mant-ATPγS | 0.136 ± 0.003 | 1.908 ± 0.012 | 14.03 |
| $\rho^{P167L}$-Psu vs. mant-ATPγS | 0.023 ± 0.001 | 0.048 ± 0.001 | 2.05 |

Increasing concentrations of Psu did not affect $dC_{15}$-FAM binding at the PBSes of either ρ variant (Fig. 6a, b), consistent with our cryoEM structures revealing unobstructed PBSes in the Psu-bridged ρ complexes. Similarly, Psu inhibited the binding of $rU_{12}$-FAM to the ρ SBS only very weakly (Fig. 6c; $IC_{50}$ = 299 μM). In contrast, the prior addition of Psu to $\rho^{P167L}$-$dC_{15}$ led to reduced $rU_{12}$-FAM binding at the SBS (Fig. 6d; $IC_{50}$ = 2.8 μM), while Psu failed to displace $rU_{12}$-FAM from pre-formed $\rho^{P167L}$-$dC_{15}$-$rU_{12}$-FAM complexes (Fig. 6e). Together with the effect of Psu on RNA-stimulated ρ ATPase, these observations suggest that

while Psu stabilizes the open conformation of $\rho^{wt}$ or $\rho^{P167L}$, it still allows binding of RNA at the center of the open $\rho^{wt}$ complexes, but inhibits RNA binding to the center of open $\rho^{P167L}$ complexes.

## Psu variants defective in ρ binding exhibit anti-termination defects in vivo

To test the importance of interactions observed in our ρ-Psu complex structures in cells, we monitored the effects of selected ρ and Psu variants under in vivo conditions. Apart from interaction-deficient

**Fig. 5 | Effects of Psu on nucleotide binding by ρ. a** Time traces monitoring wt ρ (left), $\rho^{P167L}$ (middle) or $\rho^{P167L}$-Psu (right) binding to increasing concentrations of mant-ATPγS (0.5 to 3 μM; colored from light grey to black), recorded via stopped-flow/FRET measurements. Data represent means, $n$ = 3–6. Data were fit to a single-exponential equation; $A = A_O + (A_{max} - A_O)*(1\text{-}exp(\text{-}k_{app}*t))$; $A$, fluorescence at time $t$; $A_{max}$, final signal; $A_O$, initial fluorescence signal; $k_{app}$, characteristic time constant. Source data are provided as a Source Data file. **b** Linear regression ($k_{app} = k_I$[mant-ATPγS] $+ k_{-1}$) of the apparent rate constants, $k_{app}$, at increasing mant-ATPγS concentrations, derived by the single exponential fitting of the time-traces depicted in (**a**). The association ($k_{on}$) and dissociation ($k_{off}$) rate constants were derived from the slopes and y-intercepts. $K_d$'s were calculated as the ratios of the $k_{off}$ and $k_{on}$ values. Source data are provided as a Source Data file. **c** Time traces monitoring wt ρ (left) or ρ/Psu (middle) binding to mant-ATPγS, and of $\rho^{wt}$-mant-ATPγS binding to Psu (right), recorded via stopped-flow/FRET measurements. Data represent means, $n$ = 3–6. $k_{app}$ values depicted in the graphs were derived by fitting of the data to a single-exponential equation as in (**a**) ($\rho^{wt}$ or $\rho^{wt}$/Psu binding to mant-ATPγS), or by fitting of the data to a double exponential equation ($\rho^{wt}$/Psu binding to mant-ATPγS); $A = A_O + A_1(1\text{-}exp(\text{-}k_{app1}*t)) + A_2(1\text{-}exp(\text{-}k_{app2}*t))$; $A$, fluorescence at time $t$; $A_O$, initial fluorescence signal; $A_1$ and $A_2$, amplitudes of the signal change; $k_{app1}$ and $k_{app2}$, characteristic time constants. Source data are provided as a Source Data file. **d** Time traces monitoring the dissociation (replacement with ATPγS) of mant-ATPγS from $\rho^{wt}$-mant-ATPγS-Psu (left) or $\rho^{P167L}$-mant-ATPγS-Psu (right), recorded via stopped-flow/FRET measurements. Data represent means, $n$ = 3–6. $k_{app}$ values depicted in the graphs were derived by fitting of the data to a single-exponential equation; $A = A_{min} + (A_O\text{-}A_{min})exp(\text{-}k_{app}*t)$; $A$, fluorescence at time $t$; $A_{min}$, final signal; $A_O$, initial fluorescence signal; $k_{app}$, characteristic time constant. Source data are provided as a Source Data file.

$Psu^{R43A}$ and $Psu^{D49R}$ (Fig. 3d, g), we constructed $Psu^{K180A}$, in which hydrogen bonds of K180 with E369 and T372 of ρ are abolished (Fig. 3e). We also generated $\rho^{Y197A}$, in which the cation-π stacking of Y197 with R43 of Psu is not possible (Fig. 3d). Selected, previously investigated Psu and ρ variants[38,39,44] served as controls: $Psu^{E56K}$, which cannot sustain an interaction with the backbone amide of Q374 (Fig. 3e); $Psu^{F169V}$, in which the proper positioning of helix α7 is likely compromised; and $\rho^{R146E}$, in which a salt bridge to Psu D49 is destroyed (Fig. 3d).

To monitor potential interaction defects of ρ or Psu variants in vivo, we co-produced ρ variants and N-terminally His$_6$-tagged Psu variants in the *E. coli* BL21(DE3) strain, prepared cell lysates of mid-log phase cultures, and conducted pull-down assays using Ni$^{2+}$-NTA beads, as described previously[38,39]. About 45% of ρ was pulled down via $Psu^{wt}$, while all other tested Psu variants strongly reduced co-precipitation (<8%; Fig. 7a). Thus, all Psu variants expected to be defective in ρ binding based on our structures indeed failed to bind ρ efficiently in vivo. Likewise, only about 5% or about 25% of $\rho^{R146E}$ or $\rho^{Y197A}$, respectively, were co-precipitated by $Psu^{wt}$ (Fig. 7b), consistent with ρ-Psu contacts observed in our structures.

Over-production of $Psu^{wt}$ or of Psu variants that bind and inhibit ρ leads to severe growth and ρ-dependent termination defects[39,40]. We, therefore, produced $Psu^{wt}$ and variants from an IPTG-inducible $P_{tac}$ promoter in *E. coli*, and spotted serial dilutions of saturated cultures in the absence or presence of 50 μM or 100 μM IPTG. Production of $Psu^{wt}$ caused severe growth defects, whereas the growth of most strains that produced Psu variants defective in ρ binding was unaffected (Fig. 7c). Only $Psu^{R43A}$ production led to mild growth defects in the presence of 100 μM IPTG (Fig. 7c), consistent with only partial disruption of the $\rho^{P167L}$ interaction in analytical SEC (Fig. 3g) and partial inhibition of ρ ATPase activity by this variant in vitro (Fig. 4b)

Next, we co-transformed the MG1655 strain that harbors a chromosomal deletion of the *rho* gene with the $P_{tac}$-$psu^{wt}$ plasmid and plasmids producing ρ variants. Upon induction of $Psu^{wt}$ by 50 μM IPTG, strains producing $\rho^{R146E}$ or $\rho^{Y197A}$ showed no or reduced growth defects, respectively (Fig. 7d). These results are fully in line with the lost or partially reduced interactions of $Psu^{wt}$ with $\rho^{R146E}$ or $\rho^{Y197A}$ suggested by our structures and observed in pull-down assays (Fig. 7b).

We then investigated the effects on ρ-dependent termination in vivo using a reporter system in which the *lacZ* gene was positioned downstream of a strong ρ-dependent terminator, $\lambda t_{R1}$ (Fig. 8a, top). As β-galactosidase production from the reporter depends on the suppression of ρ-dependent termination, colonies will appear white or pale blue on LB-X-gal plates if ρ is functional, and will appear blue-green if ρ function is inhibited. When $Psu^{wt}$ was expressed in the reporter strain, colonies appeared blue-green, indicating inhibition of ρ. Consistent with partial defects in ρ binding observed in vitro (Fig. 3g) and in pull-down assays (Fig. 7a), $Psu^{R43A}$-producing colonies were light blue, whereas production of all Psu variants defective in ρ

binding led to white or pale blue colonies. Quantification of β-galactosidase activity in extracts of the above strains corroborated these observations (Fig. 8a, bottom).

Finally, *E. coli* cells with the *lacZ* reporter were transformed with plasmids expressing ρ or variants, and subsequently, the chromosomal *rho* gene was deleted (Fig. 8b, top). The resulting strains were then transformed with plasmids encoding $Psu^{wt}$ (or empty plasmid as control), and the colonies were streaked on LB-X-gal plates. Colonies producing ρ in the presence of $Psu^{wt}$ appeared blue-green, indicating $Psu^{wt}$-mediated inhibition of ρ-dependent termination. In contrast, colonies producing $\rho^{Y197A}$ or $\rho^{R146E}$, defective in $Psu^{wt}$ binding based on our structures, appeared pale blue, consistent with these ρ variants being partially resistant to inhibition by $Psu^{wt}$. Again, these observations were confirmed by quantifying β-galactosidase activity in extracts of the strains (Fig. 8b, bottom).

## Discussion

P4 is a paradigmatic phage satellite that relies on its helper phage, P2, to produce functional virions. The P4-encoded Psu plays two vital roles in P4 infection: it stabilizes the Sid-hijacked P2 capsid proteins to selectively package the P4 DNA[32], and it inhibits ρ-mediated attenuation of phage genes[30]. Here, we elucidated the structural basis and molecular mechanisms underlying the Psu-mediated inhibition of ρ. Our cryoEM structures revealed that Psu instigates hyper-oligomerization of ρ on two levels. First, multiple Psu dimers cross-strut two open ρ complexes; and second, lower-level ρ-Psu complexes can incorporate additional ρ subunits, expanding ρ to at least the nonamer stage, in part aided by the concomitant incorporation of additional Psu dimers.

During capsid morphogenesis, two P4 proteins, Psu and its distant paralog Sid, remodel the P2 capsid[32]; the Psu/Sid-modified capsid can only accommodate a much smaller P4 genome, one-third the length of P2 (11 kb vs. 33 kb). In a high-resolution reconstruction of the P4 pro-capsid, Sid dimers are seen bound across hexamers of gpN, the P2 capsid protein[48]. A low-resolution structure of the mature P4 capsid, in which Psu replaces Sid, suggested that the structurally similar Psu dimer likewise caps gpN hexamers[49]. Guided by the Psu-gpN interaction as well as by mutational and cross-linking studies, a model of a ρ-Psu complex has been proposed, in which a Psu dimer also caps a closed ρ ring[38]. Our cryoEM structures of ρ-Psu complexes revealed a decisively different mode of complex formation, fully in line with previous (Supplementary Table 2) and present mutational analyses of the ρ-Psu interaction, the estimated ~1:1 subunit stoichiometry in ρ-Psu complexes[38,40], and additional biochemical data presented here. First, Psu blocks the nucleotide-binding pockets on ρ – we show that Psu reduces the binding and release of nucleotides to and from ρ (Fig. 5). Second, ρ adopts an open, inactive conformation in all complex structures with Psu – we demonstrate that Psu efficiently inhibits ρ ATPase activity (Fig. 4). Third, in all our

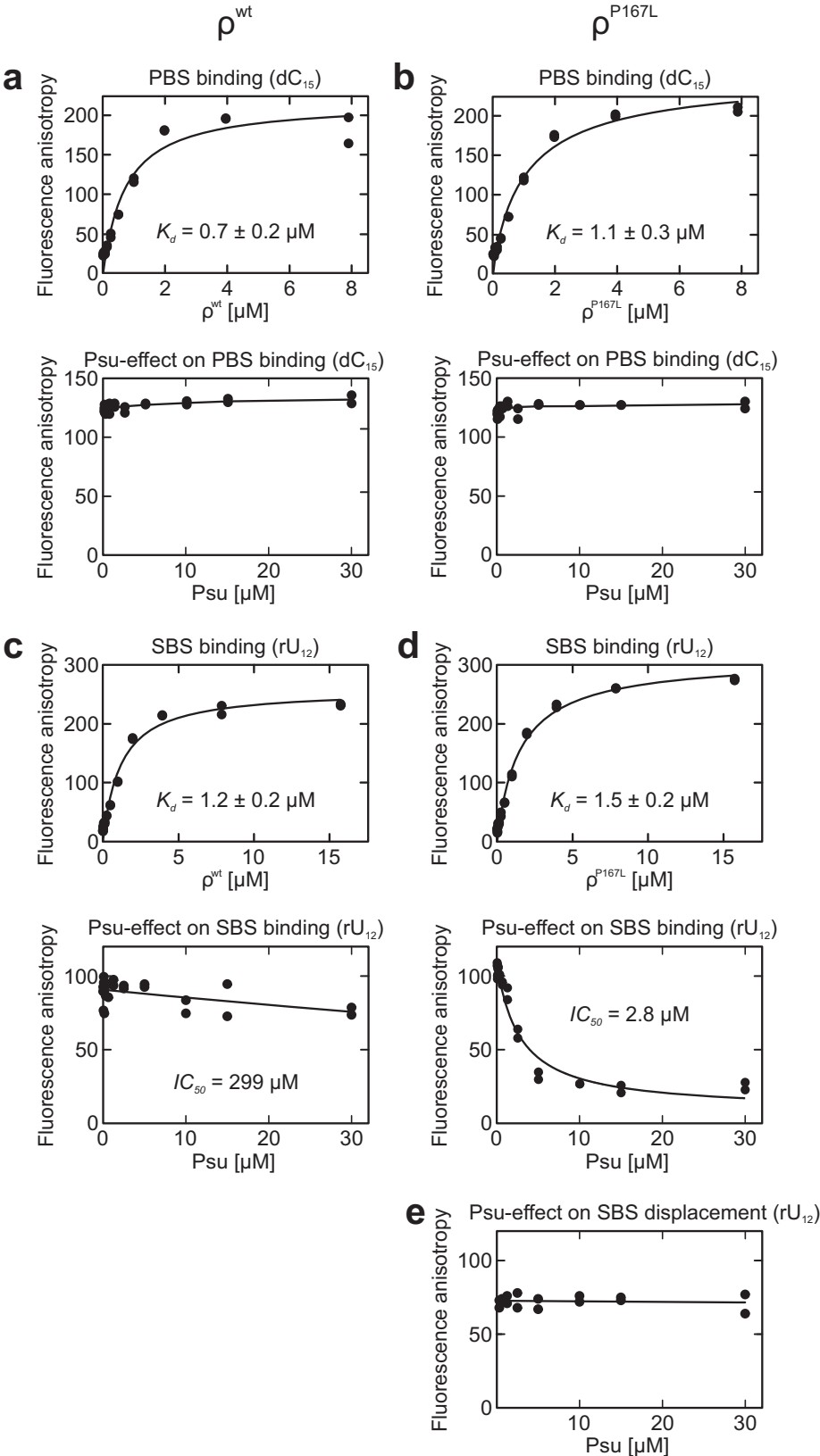

structures the NTDs of the two Psu-bridged ρ complexes are facing outwards – we show that Psu does not inhibit nucleic acid binding to the ρ PBSes (Fig. 6a). Finally, our functional analyses addressing the effects of Psu and ρ variants on the ρ-Psu interaction in cells, cell viability, and ρ-mediated termination in vivo (Figs. 7 and 8) are fully consistent with our cryoEM structures.

Although Psu inhibited the ρ ATPase, it did not prevent the binding of a small SBS ligand, $rU_{12}$-FAM, to wt ρ. SBS binding to isolated ρ is thought to be accompanied by ring closure. Presently, we cannot rigorously exclude that the wt ρ bound to Psu closes upon accommodation of RNA at the SBS; if so, Psu may remain bound to the closed rings due to conformational adjustments in the ρ-binding sites and still

**Fig. 6 | Effects of Psu on nucleic acid binding at the ρ PBS and SBS.**
**a**, **b** Fluorescence anisotropy measurements monitoring the interaction of increasing concentrations of $\rho^{wt}$ (**a**) or $\rho^{P167L}$ (**b**) with $dC_{15}$-FAM at the PBSes (top), or the effect of increasing concentrations of Psu on $\rho^{wt}$ (**a**) or $\rho^{P167L}$ (**b**) $dC_{15}$-FAM binding at the PBSes (bottom). **c**, **d** Fluorescence anisotropy measurements monitoring the interaction of increasing concentrations of $\rho^{wt}$ (**c**) or $\rho^{P167L}$ (**d**) with $rU_{12}$-FAM substrate at the SBS (top), or the effect of increasing concentrations of Psu on $\rho^{wt}$ (**c**) or $\rho^{P167L}$ (**d**) $rU_{12}$-FAM binding at the SBS (bottom). **e** Lack of Psu-mediated release of SBS-bound RNA. Data were recorded as technical duplicates, $n = 2$. Data

in (**a**–**d**, top) were fitted to a single-exponential Hill equation for $dC_{15}$-FAM or $rU_{12}$-FAM association at ρ PBSes or SBS. $A = A_{max}[\text{protein}]^h/(K_d{}^h + [\text{protein}]^h)$; in which $A_{max}$ is the fitted maximum of nucleic acid bound; $K_d$ is the dissociation constant; h is the Hill coefficient. Data in (**c**, **d** bottom) were fitted to an [inhibitor] vs. response equation: $A = A_{min} + (A_{max} - A_{min})/(1 + ([\text{inhibitor}]/IC_{50}))$; in which $A$ is the anisotropy signal at a given concentration of Psu (inhibitor); $A_{min}$ and $A_{max}$ are the fitted minimum and maximum of nucleic acid bound. Source data are provided as a Source Data file.

inhibit the ρ ATPase by restricting hydrolysis-associated ρ conformational dynamics and by covering the nucleotide binding sites. Alternatively, the small probe we used in SBS binding studies could enter the central channel of the Psu-bound open ρ rings and become immobilized without leading to ring closure. Irrespectively, a long natural *rut* site RNA would have to access the SBS via the lateral opening in the ρ rings. This lateral opening is physically obstructed in some of the observed ρ-Psu complexes (ρ-ATP/ATPγS-Psu complexes II[expanded] and III[expanded]). Also in other complexes, lateral access may be restricted due to conformational restraints imposed by Psu binding. Such conformational restraints may be more severe for $\rho^{P167L}$, which exhibits higher Psu affinity. Furthermore, the space in the central channel of the $\rho^{P167L}$-ATPγS-Psu complex II[locked] is reduced due to $\rho^{P167L}$ refolding. Both aspects could explain the more severe Psu-mediated reduction of SBS binding compared to wt ρ.

Psu-mediated restriction of access to the SBS and inhibition of ρ ring closure would interfere with the ρ translocase activity, and thus with ρ-mediated termination via the tethered tracking pathway[13]. Open and expanded ρ complexes bridged by Psu would likely also be incompatible with ρ engaging NusA/NusG-modified ECs, as recently imaged[15,16]; the inter-subunit and $\rho_1$-$\rho_6$ rise values of the open ρ rings in these pre-termination complexes are similar to those of open ρ in isolation, while they are further reduced in open ρ with bound small-molecule inhibitor, bicyclomycin (Fig. 1h). Even if ρ-Psu complexes could still engage NusA/NusG-ECs, Psu would most likely hinder the conformational changes needed to inactivate RNAP or inhibit late stages of termination/hybrid unwinding, when ρ is expected to engage and translocate on RNA[15]. We posit that the Psu-mediated blockade of the nucleotide-binding pockets of ρ contributes to the inhibition of the ρ ATPase activity. Interestingly, ATP and analogs support Psu binding to ρ (Supplementary Fig. 1a). We attribute this effect of the nucleotides to the stabilization of the Psu contact surface on the ρ CTDs and direct contacts of Psu to residues surrounding the nucleotide-binding pockets (Fig. 3f). Once bound, however, Psu inhibits the ρ ATPase activity. Thus, Psu exhibits typical features of an uncompetitive inhibitor of the ρ ATPase.

ρ activity is regulated on multiple levels. First, interactions with nucleotides, RNAs, or NusG shift the equilibrium between open and closed hexameric states and thereby regulate ρ ATPase and RNA translocase activities[10,50]. Second, RNAP-associated NusA[51,52], an RNAP-trailing ribosome[53–55], RNAP-bound elements of nascent RNA[26,27], RNAP-bound specialized transcription factors[23–25], or multi-factorial RNAP-bound RNA-protein complexes[21,22] can delay or prevent the attack of ECs by ρ. Furthermore, the RNA chaperone Hfq can counteract ρ by depositing ρ-inhibitory sRNAs on nascent transcripts[4,56]. Third, the endogenous ρ regulators, YihE and Rof, can directly bind ρ and trap it in an open hexameric state[57–59]. Results presented here underscore an additional mechanism via which ρ can be regulated, i.e., factor-dependent modulation of its oligomerization state. This mechanism capitalizes on the intrinsic oligomerization dynamics of ρ, with the hexamer representing the only state in which ρ can terminate transcription. Depending on the concentration and conditions, ρ can dissociate into non-functional, lower oligomeric states, down to the monomer level, or form dodecamers, albeit exhibiting a different configuration, as seen in complexes with Psu[60–62]. While some proteins,

such as *Vibrio cholerae* YaeO/Rof, may stabilize ρ in a hypo-oligomerized state[63], Psu stabilizes hyper-oligomerized, inactive states.

Apart from interacting proteins, some ρ orthologs have acquired additional regions or domains that facilitate ρ regulation via modulation of its oligomeric state. A prion-like domain in *Clostridium botulinum* ρ induces the formation of inactive amyloid-like aggregates, thought to underpin adaptation to stress[64]; conversely, an intrinsically disordered region (IDR) in *Bacteroides thetaiotaomicron* ρ promotes phase separation associated with increased termination activity and fitness[65]. By contrast, we have recently found that *E. coli* ρ, which lacks a prion-like domain or IDR, forms inactive large oligomers and filaments upon binding to stress-associated nucleotides, ADP and (p)ppGpp (see accompanying manuscript)[66]. We posit that condition-dependent activation or inactivation of ρ through changes in its oligomeric or aggregation state may be a widespread regulatory strategy.

Modulation of the oligomerization state as a regulatory mechanism has been observed in other molecular systems, including in eukaryotes, and is of growing interest among drug developers[67]. For example, plant or mammalian ASPL/PUX1 proteins can disassemble the essential and multi-functional AAA+ ATPase CDC48/p97[68–70], whereas biparatopic designed ankyrin repeat proteins can induce apoptosis by oligomerization of human epidermal growth factor receptor-2 in nanoscopic membrane domains[71]. The development of oligomerization-modulating compounds can be fueled by structural knowledge of the target protein in different oligomeric states dependent on a cellular interaction partner. Inhibitory peptides (shiftides) of HIV-1 integrase, for instance, have been designed based on the interacting lens epithelium-derived growth factor/p75, which shift HIV-1 integrase into an inactive, tetrameric state[72]. Results presented here could thus inspire the development of novel ρ-inhibitory substances. Peptides (and eventually peptide-derived compounds) designed based on ρ-contacting regions of Psu might not only block the nucleotide-binding pockets on ρ but may also facilitate ρ hyper-oligomerization. While short peptides or low-molecular-weight compounds will not support the dimerization of two open ρ rings akin to Psu, substances that bind to and increase the pitch of open ρ hexamers may conceivably allow expansion into inactive ρ filaments. Notably, Psu-derived, ρ-inhibiting peptides that are active in vivo have already been designed[73], but the structural basis of their modes of action remains to be explored.

Recent studies suggest that Psu may also play a hitherto unknown role in phage defense. Defense systems encoded on mobile elements are pan-genomic "guns for hire"[74] that bacteria deploy when under attack by phages[75]. In rapid exchange through horizontal transfer, these elements shape bacterial adaptation and evolution, and can inform the development of new phage therapies. P4-like elements found in enterobacterial genomes frequently carry phage defense systems[76,77], some of which can protect *E. coli* from lytic phages, including λ, P1, and T7, while not restricting P2[77].

P4 can infect diverse bacteria, prompting us to search for Psu homologs across bacteria (see Supplementary Discussion). Our analysis revealed that more than three quarters of *psu*+ genomes, including those outside of *Escherichia*, contain either a known[78] or a putative defense system inserted between the *psu* and *int* genes of P4

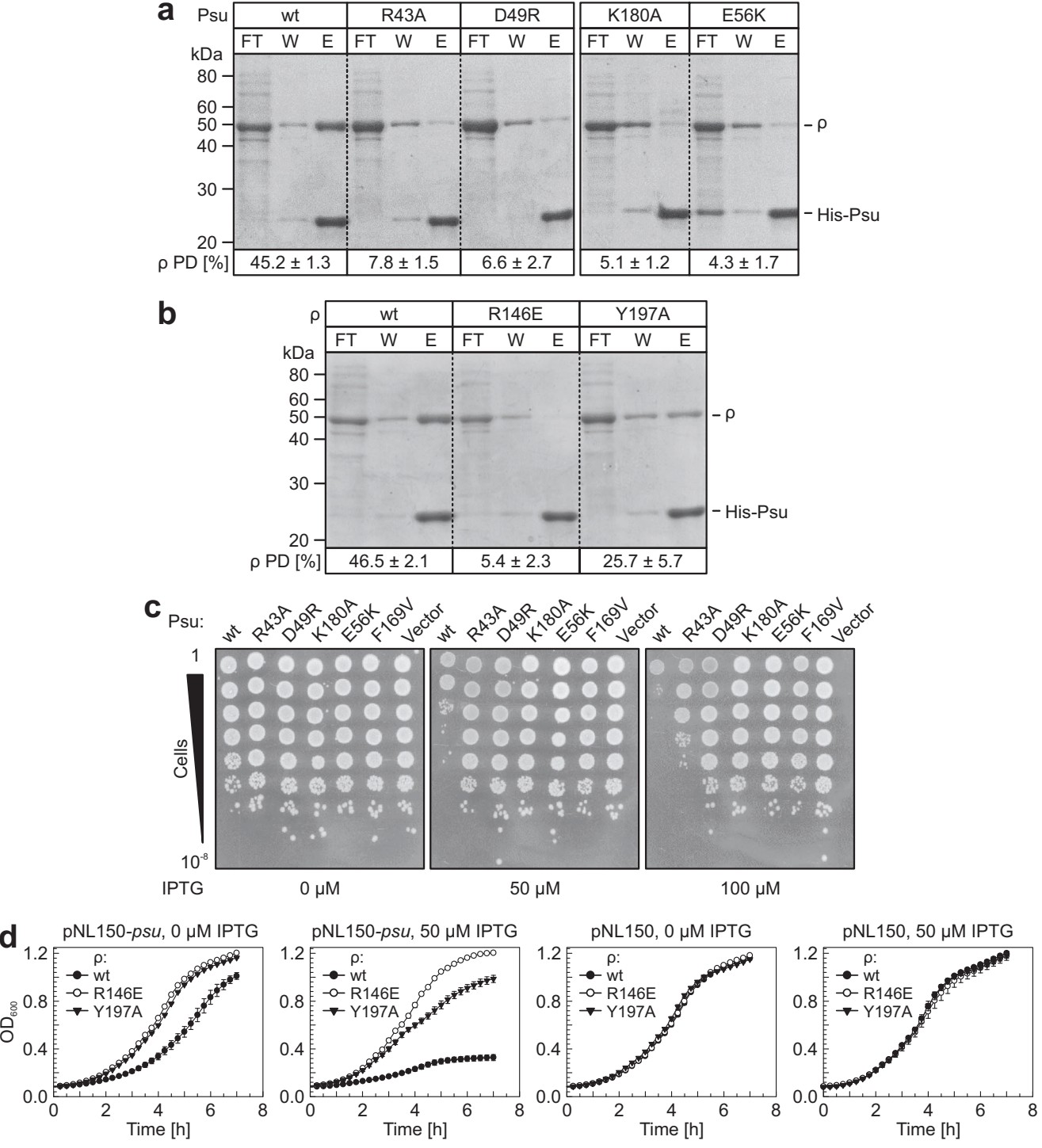

**Fig. 7 | In vivo interaction of Psu and ρ variants and effects on cell growth. a** In vivo binding defects of Psu variants. *E. coli* cell lysates after production of N-terminally His$_6$-tagged Psu variants and ρ were passed through Ni$^{2+}$-NTA beads, and the flow-through (FT), wash (W), and eluted (E) fractions were collected. The fraction of pulled-down (PD) ρ was calculated as (E / (FT + W + E)) × 100. Psu variants are indicated above the gels. Molecular mass markers are indicated on the left. Protein bands are identified on the right. Quantified data (ρ PD [%]) represent means ± SD; *n* = 3. Source data are provided as a Source Data file. **b** In vivo binding defects of ρ variants. Experiments were conducted and evaluated as in (**a**). ρ variants are indicated above the gel. Molecular mass markers are indicated on the left. Protein bands are identified on the right. Quantified data (ρ PD [%]) represent means ± SD; *n* = 3. Source data are provided as a Source Data file. **c** Growth effects elicited by Psu variants. Serial dilutions of *E. coli* overnight cultures producing Psu

variants (indicated above the plate images) were spotted on LB agar plates with different IPTG concentrations (indicated below the plate images). Serial dilutions are indicated on the left. Experiments were repeated independently at least two times with similar results. Source data are provided as a Source Data file. **d** Differential inhibition of ρ variants by Psu. *E. coli* MG1655Δ*rho*Δ*rac* producing the indicated ρ variants were transformed with a plasmid guiding production of Psu$^{wt}$ (first and second panel). As a control, strains were transformed with an empty vector (third and fourth panel). Growth curves were recorded in the absence of IPTG or in the presence of 50 μM IPTG, as indicated above the panels. Data points represent means ± SEM; *n* = 4–5 independent replicates (ρ$^{R146E}$-Psu or ρ$^{Y197A}$-Psu, 0 μM IPTG and ρ$^{Y197A}$-Psu/empty vector, 50 μM IPTG, *n* = 4; all others, *n* = 5). Source data are provided as a Source Data file.

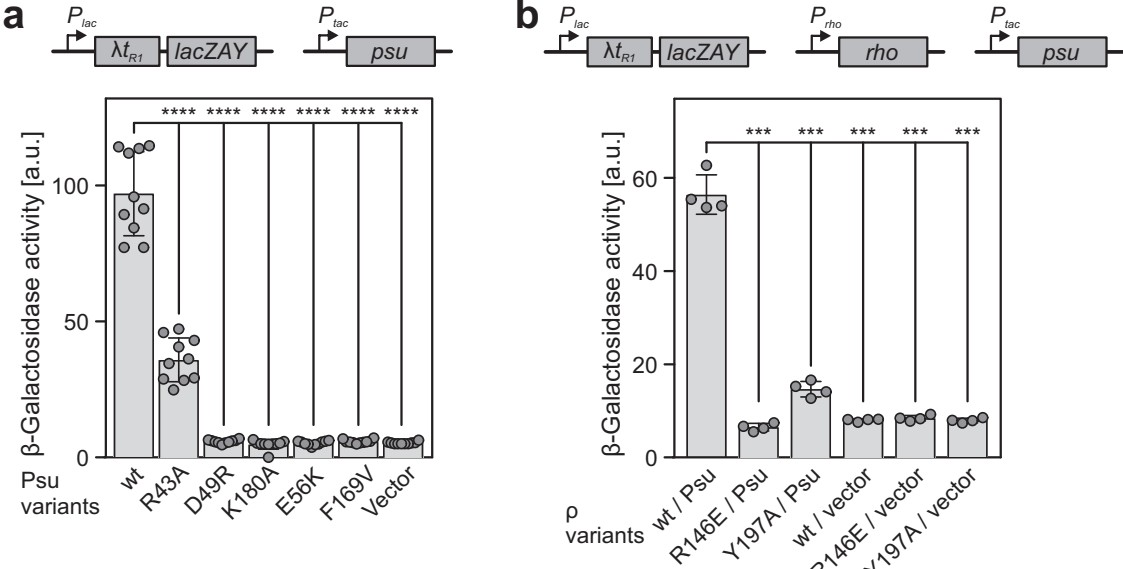

**Fig. 8 | In vivo effects of Psu and ρ variants on Psu-modulated ρ-dependent termination. a** Top, *E. coli* MG1655Δ*rac*Δ*lac* with a $P_{lac}$-$\lambda t_{R1}$-*lacZYA* reporter was transformed with plasmids guiding the production of the indicated Psu variants, or with an empty vector. Bottom, β-galactosidase activities were quantified after growing cells to $OD_{600} = 0.4$ in the presence of 15 μM IPTG. Higher β-galactosidase activities represent less efficient ρ-dependent termination at the ρ-dependent terminator, $\lambda t_{R1}$, due to absence or presence of the indicated Psu variants. Spheres, individual data points. Boxes represent means ± SD of ten biological replicates (individual colonies). ****$p < 0.0001$ (wt Psu vs. indicated Psu variants or empty vector). *p*-values were calculated via one-way analysis of variance (ANOVA). Source data are provided as a Source Data file. **b** Top, *E. coli* MG1655Δ*rho*Δ*rac*Δ*lac* with a

$P_{lac}$-$\lambda t_{R1}$-*lacZYA* reporter and producing the indicated ρ variants were further transformed with a plasmid guiding the production of Psu^wt, or with an empty vector. Bottom, β-galactosidase activities were quantified after growing cells to $OD_{600} = 0.4$ in the presence of 5 μM IPTG. Higher β-galactosidase activities represent less efficient ρ-dependent termination at the ρ-dependent terminator, $\lambda t_{R1}$, due to differential inhibition of the ρ variants by wt Psu or due to uninhibited ρ variants. Spheres, individual data points. Boxes represent means ± SD of four biological replicates (individual colonies). ***$p < 0.001$ (wt ρ/Psu vs. indicated ρ variants/Psu or indicated ρ variants/empty vector). *p*-values were calculated via one-way ANOVA. Source data are provided as a Source Data file.

(Supplementary Fig. 13). The Psu-associated defense systems are astonishingly diverse, representing 36 known classes, and are enriched in Abi modules, such as Gabija[79], retrons[80], and PD-T7-2[81]; whereas the restriction-modification systems, which dominate the bacterial defense arsenal[78], are underrepresented (Supplementary Fig. 13c).

We speculate that Psu proteins may serve as built-in anti-terminators that promote the expression of some defense genes. Many defense genes are codirectional with *psu* (Fig. 9), potentially extending the *psu* operon, which is known to be activated by Psu[30]. Notably, the molecular mechanism of Psu is distinct from those of canonical phage anti-terminators, such as phage λ N and Q proteins, which are recruited to transcribing RNAP at specific RNA or DNA elements and activate expression of genes downstream of these elements[82]. Psu binds to ρ and can thus act on any gene that is susceptible to ρ-mediated termination, possibly more efficiently when produced *in cis*.

While supplying the cell with antiviral weapons, a Psu/defense cassette also poses a threat to its very existence. The expression of Abi systems must be tightly controlled to ensure that a suicide program is triggered only when the first line of cellular defense has been breached[83]. In fact, Psu itself is a potent toxin that, if present at high levels, triggers cell death in diverse bacteria[84]. ρ could be a key part of the silencing mechanism: it limits the expression of the *psu* gene[30] and may be intrinsically resistant to Psu: P167L substitution stabilizes ρ-Psu interactions[38,43], and we traced this enhanced stability to a tendency of ρ^P167L to refold and form domain-swapped dimers at the center of ρ^P167L-Psu complexes. While ρ^P167L does not exhibit defects in laboratory assays[38], the proline at the 167-equivalent position is invariant (Supplementary Fig. 12a), perhaps to provide higher ρ stability. In *Enterobacteriaceae*, the invariance of P167 conveniently limits ρ-Psu affinity and thereby balances Psu-dependent expression of phage defense genes and cellular toxicity of Psu.

## Methods

### Recombinant protein production and purification

Primers used for molecular cloning are listed in Supplementary Table 3. Plasmids used for protein production or as templates for in vitro transcription are listed in Supplementary Table 4. Genes encoding Psu variants (R43A and D49R) were produced by the single-primer quick mutagenesis method[85]. For protein production, *E. coli* BL21 RIL cells were transformed with corresponding plasmids. ρ, ρ^P167L and Psu^wt were produced and purified as previously described[15,38,86]. Briefly, lysates containing non-tagged ρ or ρ^P167L were loaded on a heparin column equilibrated in 50 mM TRIS-HCl, pH 7.5, 50 mM NaCl, 5 % (v/v) glycerol, 1 mM DTT. The proteins were eluted with a gradient to 1 M NaCl, and the fractions containing ρ or ρ^P167L were pooled, concentrated and loaded on a Superdex 200 column for SEC in 10 mM TRIS-HCl, pH 7.5, 150 mM NaCl, 1 mM DTT. Psu^wt or variants were precipitated with 25% final concentration of ammonium sulfate for 1 h at RT, and pelleted by $15,000 \times g$ centrifugation at 4 °C for 20 min. The pellet was resuspended in 20 mM TRIS-HCl, pH 7.9, 150 mM NaCl, 5% (v/v) glycerol, 0.1 mM EDTA, 1 mM DTT and passed through heparin and MonoQ chromatography columns to clear contaminants. The flow-through was subjected to MonoS cation-exchange chromatography, eluted with a gradient to 1 M NaCl and the fractions containing Psu variants were pooled, concentrated and loaded on a on a Superdex 75 column for SEC in 20 mM TRIS-HCl, pH 8.0, 100 mM NaCl, 10% (v/v) glycerol, 0.1 mM EDTA, 1 mM DTT.

### *Rut* RNA production

A DNA template for in vitro transcription[22], harboring a T7 RNAP promoter and the sequence encoding the $\lambda t_{R1}$ *rut* site, was produced from plasmid pUC18-$\lambda t_{R1}$-*rut* (Supplementary Table 4) by PCR with a reverse primer containing two 2′-O-methylated nucleotides at the 5′-terminus, for decreasing possible 3′-end run-off products[87,88]. The produced $\lambda t_{R1}$ *rut* site

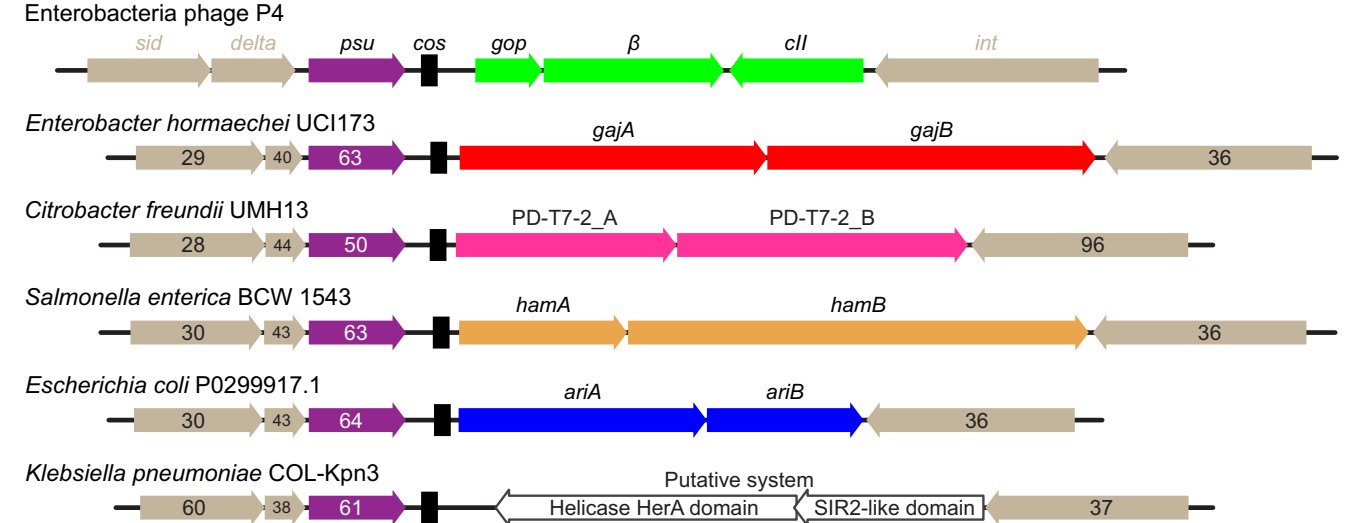

**Fig. 9 | Psu as a marker of phage defense systems.** Examples of *psu*-associated defense system loci. Numbers indicate percent amino acid residue identities to P4 *sid*, *delta*, *psu*, and *int*-encoded proteins. Delta homologs shorter than the P4 Delta protein (NP_042043.1) contain homologous regions corresponding to residues 1-40 and 120-160 of P4 Delta. The locations of *cos* sites (black boxes) were identified by multiple sequence alignments of DNA sequences within 300 bp downstream of the *psu* gene using Clustal Omega[107] and *cos* sequences from reference[108]. The known defense genes are colored (see Supplementary Fig. 13c). Some loci encode known toxin-antitoxin pairs: in Hachiman, HamA is a DNase, and HamB is a protective helicase[109], whereas in PARIS, AriB is a putative RNase and AriA is an SMC-like protein that sequesters AriB in an inactive state[110]. In other two-component systems, such as Gabija, the nuclease (GajA) and helicase (GajB) are both required for phage restriction[111]. A putative defense system with a SIR2-like domain is shown in white.

RNA was treated with DNase I and purified by strong anion-exchange (MonoQ, GE Healthcare) and size-exclusion chromatography (Superdex 75, GE Healthcare) in 10 mM HEPES-NaOH pH 7.5, 50 mM NaCl. The purity of the RNA was assessed by urea-PAGE, stained with ethidium bromide.

**Analytical size-exclusion chromatography**

Analytical SEC-based interaction analyses were performed in 10 mM TRIS-HCl, pH 7.5, 150 mM NaCl, 5 mM MgCl$_2$, 1 mM DTT. 300 pmol $\rho^{wt/P167L}$ hexamer and 1200 pmol Psu dimer (wt or variants) were mixed in a final reaction volume of 50 µl in the absence or presence of 2 mM ATP or analogs. After incubation of the mixtures at 32 °C for 10 min, the samples were loaded on a Superose 6 increase 3.2/300 analytical size exclusion column (Cytiva). 50 µl fractions were collected, analyzed by SDS-PAGE and visualized by Coomassie staining.

**Dynamic light scattering**

DLS measurements of $\rho$, $\rho^{P167L}$, Psu, and their complexes were performed using the Prometheus Panta instrument (NanoTemper). 7.4 µM of $\rho$ or $\rho^{P167L}$ hexamer were incubated alone or with a threefold molar excess of Psu dimers in 50 mM TRIS-HCl, pH 8.0, 120 mM KOAc, 5 mM Mg(OAc)$_2$, 10 µM ZnCl$_2$, 2 mM DTT for 15 min at room temperature without nucleotide or in the presence of 2 mM ATP or analogs. Isothermal DLS scans (10 acquisitions for each of three replicates, 5 seconds each, 20% LED intensity, 100% DLS laser intensity) were recorded at 15 °C in Prometheus High Sensitivity Capillaries (NanoTemper). Hydrodynamic radii (r$_H$) and polydispersity indices (PDI) were determined using the Prometheus Panta analysis software. The software employs size distribution analysis to calculate the discrete particle sizes and their proportions present within the sample and cumulants analysis to determine the average particle size present in the entire sample. Buffer viscosity base values were provided by the Prometheus Panta control software. The standard deviation around the mean r$_H$ was used to determine the PDI.

**Cryogenic electron microscopy**

$\rho$-ATP-Psu, $\rho$-ATP$\gamma$S-Psu and $\rho^{P167L}$-ATP$\gamma$S-Psu complexes were freshly prepared in 10 mM TRIS-HCl, pH 7.5, 150 mM NaCl, 5 mM MgCl$_2$ ($\rho$-ATP-Psu and $\rho$-ATP$\gamma$S-Psu), or 10 mM TRIS-HCl, pH 8.0, 120 mM KOAc, 5 mM Mg(OAc)$_2$, 5 µM ZnCl$_2$ ($\rho^{P167L}$-ATP$\gamma$S-Psu), concentrated to 5.5 mg/ml using a 100 kDa ultra-centrifugal filter (Merck) and supplemented with 1 mM ATP or ATP$\gamma$S. 3.8 µl of the samples were applied to glow-discharged holey carbon R1.2/1.3 copper grids (Quantifoil Microtools) and plunge-frozen in liquid ethane using a Vitrobot Mark IV (Thermo Fisher) equilibrated at 10 °C and 100% humidity. Micrographs were recorded on a FEI Titan Krios G3i transmission electron microscope operated at 300 kV with a Falcon 3EC detector. Movies were recorded for 40.57 s, accumulating a total electron flux of ~ 40 e⁻/Å$^2$ in counting mode at a calibrated pixel size of 0.832 Å/px distributed over 33 fractions.

**CryoEM data analysis**

All image analysis steps were done with cryoSPARC (version 3.2.2)[89]. Movie alignment was done with patch motion correction generating Fourier-cropped micrographs (pixel size 1.664 Å/px), and CTF estimation was conducted by Patch CTF. Class averages of manually selected particle images were used to generate an initial template for reference-based particle picking from 6409/5986/2723 micrographs ($\rho$-ATP-Psu/$\rho$-ATP$\gamma$S-Psu/$\rho^{P167L}$-ATP$\gamma$S-Psu). 966,398/1,815,462/823,078 particle images were extracted with a box size of 224 px and Fourier-cropped to 112 px for initial analysis. Reference-free 2D classification was used to select 560,412/747,325/763,878 particle images for further analysis. Ab initio reconstruction using a small subset of particles was conducted to generate an initial 3D reference for subsequent classification by 3D heterogeneous refinement or 3D variability analysis. Local motion correction was applied to extract particle images at full spatial resolution with a box size of 448 px followed by global and local CTF refinement. Due to considerable structural and compositional flexibility of the complexes, classification was not straightforward. Instead, similar structures appeared multiple times during classification, which were then combined for final homogeneous refinement. For the $\rho$-ATP-Psu sample, 17,296 and 64,053 particle images were selected for final non-uniform (NU) refinement, yielding reconstructions at global resolutions of 3.8 Å (complex II) and 3.6 Å (complex II$^{expanded}$), respectively. For the $\rho$-ATP$\gamma$S-Psu sample, 73,056 and 15,407 particle images

were selected for final NU refinement, yielding reconstructions at global resolutions of 3.65 Å (complex III) and 4.25 Å (complex III$^{expanded}$), respectively. For the $\rho^{P167L}$-ATPγS-Psu sample, 273,935 and 317,635 particle images were selected for final NU refinement, yielding reconstructions at global resolutions of 3.09 Å (complex II) and 2.95 Å (complex II$^{locked}$), respectively.

## Model building, refinement and analysis

Crystal structures of $\rho$ (PDB ID: 1PV4)[90] and Psu (PDB ID: 3RX6)[86] were manually placed in the cryoEM reconstructions and each protomer was adjusted by rigid body fitting and segmental real-space refinement using Coot (version 0.9.6)[91,92]. For higher-oligomeric complexes, additional $\rho$ subunits and Psu dimers were added to unoccupied regions of the cryoEM reconstructions and adjusted as above. The models were refined by iterative rounds of real space refinement in PHENIX (version 1.20_4459)[93,94] and manual adjustment in Coot. The structural models were evaluated with MolProbity (version 4.5.1)[95]. Structure figures were prepared with PyMOL (version 2.4.0, Schrödinger) and ChimeraX[96].

Helical parameters for $\rho$-Psu complexes were calculated with PyMOL, using the draw_rotation_axis.py script available from the PyMOL script repository (https://github.com/Pymol-Scripts/Pymol-script-repo). The axis was drawn for the inner-most $\rho$ hexamers in the respective complexes. The intersubunit and $\rho_1$-$\rho_6$ rise values were derived from the length of the corresponding translational vectors along the rotation axis.

## Differential scanning fluorimetry

Differential scanning fluorimetry was conducted in a 96-well plate in a plate reader combined with a Mx3005P thermocycler (Stratgene). $\rho$ variants were prepared at a final concentration of 2 µM in 10 mM TRIS-HCl, pH 7.5, 150 mM NaCl, 1 mM DTT, supplemented with 1x SYPRO orange in a final volume of 20 µl. The temperature was increased at a rate of 1 °C/min from 25 °C to 95 °C, and the fluorescence emission was monitored at each step. The fluorescence intensity was plotted as a function of temperature. The thermal melting temperature was determined from the melting profile as the temperature at which the fluorescence intensity reached 50% of the highest fluorescence intensity.

## ATPase assays

TLC-based ATPase assays were performed using [α-$^{32}$P]ATP (Hartmann Analytic). To quantify RNA-stimulated ATPase activity, 100 nM $\rho$ variants, in isolation or supplemented with 1 µM or 5 µM Psu, were mixed with 20 µM *rut* RNA in 10 mM TRIS-HCl, pH 7.5, 100 mM NaCl, 5 mM MgCl$_2$, 1 mM DTT. The mixtures were incubated with 1 mM ATP, supplemented with [α-$^{32}$P]ATP, at 37 °C for up to 2 h. 2 µl of the samples were withdrawn at selected time points and the reactions were quenched with 6 µl of 100 mM EDTA, pH 8.0. 3 × 1 µl of samples were spotted on a PEI-cellulose TLC membrane and chromatographed with 1 M acetic acid, 0.5 M LiCl, 20% (v/v) ethanol. The corresponding ATP and ADP spots were visualized using a Storm 860 phosphorimager (GMI) and quantified using ImageQuant software (version 5.2; Cytiva). To test the effect of Psu variants on $\rho$ ATPase activity, increasing concentrations of Psu variants, up to 15 µM, were mixed with 100 nM $\rho$ variants, and subsequently with 20 µM *rut* RNA, in 10 mM TRIS-HCl, pH 7.5, 100 mM NaCl, 5 mM MgCl$_2$, 1 mM DTT. The mixtures were incubated with 1 mM ATP, supplemented with [α-$^{32}$P]ATP, at 37 °C for 1 h. Subsequently, the experiments were conducted as above. Data from two biological replicates were plotted and analyzed using Prism software (version 9.0.2; GraphPad). $\rho$ or $\rho^{P167L}$ ATPase activity was calculated as the percentage of hydrolyzed ATP over time, by fitting quantified data to the equation $A = A_O + (A_{max}-A_O)*(1-\exp(-k*t))$, in which $A_O$ is the fraction of ATP hydrolyzed at time zero; $A_{max}$ is the fraction of ATP hydrolyzed at infinite time; and $k$ is the rate constant.

To quantify the effect of Psu on $\rho$ or $\rho^{P167L}$ ATPase activity, data were fitted to an inhibitor-vs.-response function: $A = A_{min} + (A_{max} - A_{min})/(1 + ([inhibitor]/IC_{50}))$, in which $A$ is the fraction of ATP hydrolyzed at a given Psu (inhibitor) concentration; and $A_{min}$ and $A_{max}$ are the fitted minimum and maximum fractions of ATP hydrolyzed.

## Stopped-flow/FRET-based analysis of nucleotide binding

Nucleotide binding studies were conducted on an SX-20MV stopped-flow/fluorescence spectrometer (Applied Photophysics). Tryptophan residues near the ATP binding pockets served as FRET donors and were excited at 280 nm. A 420 nm cut-off filter was used for detecting the emission by the acceptor, mant-ATPγS. All experiments were performed with 100 nM $\rho$ or $\rho^{P167L}$, 500 nM Psu and/or 2 µM mant-ATPγS in 10 mM TRIS-HCl, pH 7.5, 150 mM NaCl, 5 mM MgCl$_2$. For quantifying nucleotide association, data was acquired at 18 °C or 37 °C by fast mixing of equal volumes (60 µl) of the reactants (syringe 1, $\rho$ or $\rho^{P167L}$ in isolation or in the presence of Psu; syringe 2, mant-ATPγS). FRET signals were recorded for up to 30 min. For detecting the binding of Psu to $\rho$ or $\rho^{P167L}$ pre-bound to mant-ATPγS, the experiment was performed at 37 °C by rapidly mixing equal volumes (60 µl) of the reactants (syringe 1, $\rho$ or $\rho^{P167L}$ and mant-ATPγS; syringe 2, Psu) and the monitoring fluorescence changes over 30 min. Data were acquired in logarithmic sampling mode and visualized using the Pro-Data Viewer software package (version 2.5; Applied Photophysics). The final curves were obtained by averaging 3 to 6 individual traces and analyzed in Prism software (version 9.0.2; GraphPad). Unless stated otherwise, data were fitted to a single exponential function: $A = A_O + (A_{max} - A_O)*(1-\exp(-k_{app}*t))$; $A$, fluorescence at time $t$; $A_{max}$, final signal; $A_O$, initial fluorescence signal; $k_{app}$, characteristic time constant. For $\rho$ binding to mant-ATPγS and the subsequent association of Psu, a double exponential function was used: $A = A_O + A_1(1-\exp(-k_{app1}*t)) + A_2(1-\exp(-k_{app2}*t))$; $A$, fluorescence at time $t$; $A_O$, initial fluorescence signal; $A_1$ and $A_2$, amplitudes of the signal change; $k_{app1}$ and $k_{app2}$, characteristic time constants. Dependencies of the apparent rate constants on nucleotide concentration were fitted by a linear equation, $k_{app} = k_1[mant-ATPγS] + k_{-1}$; $k_1$, nucleotide association rate constant (derived from the slope); $k_{-1}$, nucleotide dissociation rate constant (derived from the Y-axis intercept). For mant-ATPγS dissociation, data were fitted to a single exponential function: $A = A_{min} + (A_O - A_{min})\exp(-k_{app}*t)$; $A$, fluorescence at time $t$; $A_{min}$, final signal; $A_O$, initial fluorescence signal; $k_{app}$, characteristic time constant.

## Nucleic acid binding assays

Nucleic acid binding to $\rho$ or $\rho^{P167L}$ PBS or SBS was tested via fluorescence depolarization-based assays[46]. For PBS binding, 5 µM FAM-labeled dC$_{15}$ oligo (Eurofins) were mixed with increasing amounts of $\rho$ or $\rho^{P167L}$ (0 to 8 µM final hexamer concentration) in 20 mM HEPES pH 7.5, 150 mM KCl, 5% (v/v) glycerol, 5 mM MgCl$_2$, 0.5 mM TCEP. For SBS binding, $\rho$ or $\rho^{P167L}$ PBSes were first saturated with 10 µM non-labeled dC$_{15}$. Increasing amounts of PBS-blocked $\rho$ or $\rho^{P167L}$ (0 to 16 µM final hexamer concentration) were then mixed with 2 mM ADP-BeF$_3$ and 2 µM FAM-labeled rU$_{12}$ oligo (Eurofins). For examining the effect of Psu on nucleic acid binding to the $\rho$ or $\rho^{P167L}$ PBS or SBS, 1 µM of $\rho$ or $\rho^{P167L}$ hexamers were mixed with increasing amounts of Psu (0 to 30 µM final dimer concentration). Subsequently, experiments were conducted as above. The fluorescence anisotropy was recorded in OptiPlateTM 384-well plates (PerkinElmer) using a Spark Multi-mode Microplate reader (Tecan; excitation wavelength, 485 nm; detected emission wavelength, 530 nm). Two technical replicates were averaged for each sample and the data were analyzed with Prism software (version 9.0.2; GraphPad). To quantify $\rho$ or $\rho^{P167L}$ PBS or SBS binding, data were fitted to a single exponential Hill function; $A = A_{max}[protein]^h/(K_d^h + [protein]^h)$; in which $A_{max}$ is the fitted maximum of nucleic acid bound; $K_d$ is the dissociation constant; and h is the Hill coefficient. For quantification of the effects of Psu on $\rho$ or

$\rho^{P167L}$ PBS or SBS binding, data were fitted to an inhibitor vs. response function: $A = A_{min} + (A_{max} - A_{min}) / (1 + ([\text{inhibitor}]/IC_{50}))$; in which $A$ is the anisotropy signal at a given concentration of Psu (inhibitor); and $A_{min}$ and $A_{max}$ are the fitted minimum and maximum of nucleic acid bound.

## Generation of *rho* and *psu* mutants for in vivo assays

A low-copy-number pCL1920 plasmid guiding the production of $\rho^{Y197A}$ was constructed by site-directed mutagenesis of a pCL1920 plasmid encoding $\rho$ (pRS649)[97]. pNL150 plasmids guiding the production of Psu[R43A], Psu[D49R] or Psu[K180A] were constructed by site-directed mutagenesis of a pNL150 plasmid encoding Psu[wt] (pRS1117)[44]. Mutations were confirmed by sequencing.

## Bacterial growth assays

To monitor possible growth defects of *rho* mutants in the absence and presence of Psu, *E. coli* RS1309 (MG1655$\Delta$*rho*$\Delta$*rac*) strain was transformed with pCL1920 plasmids expressing wt or mutant *rho* and were plated on LB in the absence of IPTG. After removal of the shelter plasmid expressing wt *rho*, the strains were transformed with empty pNL150 or with pNL150 expressing wt *psu*. The transformed colonies were streaked on LB agar plates supplemented with 0 μM or 50 μM IPTG and incubated at 37 °C. Serial dilutions of the overnight cultures of each transformed strain were spotted onto LB agar plates supplemented with 0, 50 or 100 μM IPTG.

To monitor possible growth defects of the cells expressing *psu* mutants, *E. coli* MG1655 (RS1263) strain was transformed with pNL150 plasmids expressing either wt or mutant *psu*. The transformed colonies were streaked on LB agar plates supplemented with 0 μM or 50 μM IPTG.

To record growth curves, overnight cultures of the transformed strains were cultured in 96-well microtiter plates in the absence of IPTG or in the presence of 50 μM IPTG. The growth was monitored by measuring the absorbance at 660 nm using a Spectramax M5 plate reader. The curves were plotted using SigmaPlot software (version 13).

## Pull-down assays

The pET28b (*kan*[R]) plasmids guiding the production of Psu variants with an N-terminal His6-tag, and pET21b (*amp*[R]) plasmids guiding the production of $\rho$ variants were co-transformed in *E. coli* BL21(DE3) cells. The transformed colonies were used to inoculate 5 ml of LB medium, and cultures were grown at 37 °C for ~3 h. The 5 ml cultures were then added to 100 ml LB medium and further incubated until the $OD_{600}$ reached ~0.3. The cultures were then induced with 0.1 mM IPTG for 3 h. Cells were then lysed in 100 mM $NaH_2PO_4$, pH 7.0, 100 mM NaCl, 10 mM imidazole, 1 mg/ml lysozyme, 10 mg/ml PMSF. The lysates were passed through $Ni^{2+}$-NTA columns (Qiagen), washed with 100 mM $NaH_2PO_4$, pH 7.0, 100 mM NaCl, 20 mM imidazole, and the bound protein was eluted with 100 mM $NaH_2PO_4$, pH 7.0, 100 mM NaCl, 500 mM imidazole. Identical volumes of all buffers were used in each step. Identical volumes of the flow-through (FT), wash (W) and eluted fractions (E) were separated via SDS-PAGE. The amounts of $\rho$ and Psu proteins in each of the FT, W and E fractions were quantified by ImageQuant (version 5.2; Cytiva) of the corresponding bands of the Coomassie-stained gels.

## In vivo ρ-dependent transcription termination assays

A *lacZ* reporter system was used for studying in vivo ρ-dependent transcription termination. *E. coli* RS2047 strain, containing the reporter cassette, $P_{lac}$-λ$t_{RI}$-*lacZYA*, in the form of a λRS45 lysogen, was transformed with pCL1920 plasmids that guide the production of ρ variants. Subsequently, cells were transduced with *rho::kan*[R] lysate to remove the chromosomal copy of *rho*. Finally, the modified strains were transformed with an empty pNL150 vector or with pNL150 expressing *psu*. Transformed colonies were streaked on LB-X-gal agar plates supplemented with 0 or 5 μM IPTG, and were incubated at 37 °C for 14 to 16 h.

To study the effect of *psu* mutants on ρ-dependent transcription termination in vivo, the RS2047 strain was transformed with empty pNL150 or with pNL150 expressing wt or mutant *psu*. The transformed colonies were streaked on LB-X-gal agar plates supplemented with 0 or 15 μM IPTG and were incubated at 37 °C for 12 to 14 h.

For quantification of β-galactosidase activity, transformed colonies were inoculated in the presence of 5 μM or 15 μM IPTG (as above) and allowed to growth until $OD_{600}$ 0.4. β-galactosidase assays were performed according to standard procedure.

## Compiling Psu sequences

Although a previous study[76] revealed that Psu homologs are limited to three genera, *Escherichia*, *Salmonella*, and *Klebsiella*, a broad specificity of Psu[84] suggests that its homologs may be more widely distributed. To explore this possibility, we sought to identify as many Psu homologs as possible in two steps: (1) searching UniProtKB (~249 million sequences as of Nov. 14th, 2024) and NCBI RefSeq non-redundant protein (~365 million sequences) databases using phage P4 Psu reference sequence (NCBI ID: NP_042044.1) as a query, and 2) validating the hits as true Psu homologs. First, we performed a jackhmmer search against UniProtKB (v.2021_04) on HmmerWeb (v. 2.41.2)[98] (https://www.ebi.ac.uk/Tools/hmmer). Jackhmmer attempts to find more homologs by building a new HMM at each iteration; in our study, the total number of retrieved sequences did not change after seven iterations, suggesting convergence. Next, we performed a blastp search against the RefSeq non-redundant protein database (https://blast.ncbi.nlm.nih.gov/Blast.cgi). In an attempt to find more viral Psu homologs, we also used a blastp search limited to the Virus superkingdom (Taxonomy ID: 10239) in the NCBI non-redundant protein database, a superset of the RefSeq non-redundant protein database that contains ~840 million sequences. An *E*-value of $10^{-4}$ was used during searches of large databases.

All sequences were then downloaded from NCBI. An all-vs-all blastp was performed using BLAST+ (v. 2.9.0)[99] with an *E*-value of $10^{-9}$. A Psu similarity network was built based on *E*-value and visualized in Cytoscape (v. 3.9.1)[100]. Sequences from two Psu-like groups in the network (Supplementary Fig. 13) were fed into Batch CD-Search[101] to confirm the presence of the Psu superfamily (NCBI ID: cl06476). Sequences with >80% coverage of the Psu superfamily and sequences with coverage <80% but lacking "incomplete" flags were selected. Finally, 2624 Psu sequences were kept (Supplementary Data 1a). Among them, 2523 of Psu homologs were obtained from the NCBI database and an additional 101 sequences were retrieved from the UniProtKB database. NCBI taxonomy ID was first assigned to the Psu sequences using accession2taxid file (https://ftp.ncbi.nlm.nih.gov/pub/taxonomy/accession2taxid). Then NCBI taxonomy was replaced with GTDB taxonomy[102]. As the reference sequences are not organism-specific, the taxonomy was assigned to the genus level.

## Psu conservation

The compiled Psu sequences from Pseudomonadota were clustered at 90% (-c 0.9 option) using cd-hit (v. 4.8.1)[103]. Then sequences from the virus were added into the pool. A total of 124 sequences were aligned using MUSCLE (v. 5.1) with default settings (https://drive5.com/muscle5). The multiple sequence alignment of Psu was indexed according to Psu reference sequence before generating sequence logo using WebLogo (v. 3.7.8)[104].

## Identification of phage defense systems

The accession numbers of ten neighboring genes on both sides of *psu* were fetched by TREND[105] and the sequences were downloaded from NCBI. A total of 1163 *psu*-containing genomes were investigated. The presence of Enterobacteria phage P4-like genes among the neighboring genes was determined using blastp. The proteins from phage P4 reference genome (NCBI ID: GCF_000846325.1) were used as queries, and *E*-

value threshold was set to $10^{-5}$. To find known defense system, all neighboring proteins were submitted to DefenseFinder[78]. The results were checked manually. Gene clusters located between *psu* and *int* genes that were not identified by DefenseFinder were considered putative defense systems (Supplementary Data 1d) and genes from putative defense systems were annotated (Supplementary Data 1f, g) by searching against Pfam-A database (v. 35.0) using hmmscan (v. 3.2.1)[106].

## Reporting summary

Further information on research design is available in the Nature Portfolio Reporting Summary linked to this article.

## Data availability

CryoEM reconstructions have been deposited in the Electron Microscopy Data Bank (https://www.ebi.ac.uk/pdbe/emdb) under accession codes EMD-51235 (ρ-ATP-Psu complex II), EMD-51236 (ρ-ATP-Psu complex II$^{expanded}$), EMD-17637 (ρ-ATPγS-Psu complex III), EMD-17639 (ρ-ATPγS-Psu complex III$^{expanded}$), EMD-17640 (ρ$^{P167L}$-ATPγS-Psu complex II), EMD-17641 (ρ$^{P167L}$-ATPγS-Psu complex II$^{locked}$) and EMD-51237 (ρ$^{P167L}$-ATPγS). Structure coordinates have been deposited in the RCSB Protein Data Bank (https://www.rcsb.org) with accession codes 9GCS (ρ-ATP-Psu complex II), 9GCT (ρ-ATP-Psu complex II$^{expanded}$), 8PEU (ρ-ATPγS-Psu complex III), 8PEW (ρ-ATPγS-Psu complex III$^{expanded}$), 8PEX (ρ$^{P167L}$-ATPγS-Psu complex II), 8PEY (ρ$^{P167L}$-ATPγS-Psu complex II$^{locked}$) and 9GCU (ρ$^{P167L}$-ATPγS). All other data are contained in the manuscript or the Supplementary Information. Source data are provided with this paper. Structure coordinates used in this study are available from the RCSB Protein Data Bank (https://www.rcsb.org) under accession codes 1PV4, 1XPO, 3RX6, 5JJI, 6WA8, 6XAS and 6Z9P. Source data are provided with this paper.

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

## Acknowledgements

We thank Benedikt Kirmayer and Jörg Bürger, Freie Universität Berlin, Research Center of Electron Microscopy and Core Facility BioSupraMol, for help with cryoEM sample preparation and Maxine Gripberg for help with preparation of wt ρ. We acknowledge the assistance of the core facility BioSupraMol supported by the Deutsche Forschungsgemeinschaft in electron microscopic analyses. This work was supported by grants from the Deutsche Forschungsgemeinschaft (INST 130/1064-1 FUGG to Freie Universität Berlin; GRK 2473 "Bioactive Peptides", project number 392923329, to M.C.W.; WA 1126/11-1, project number 433623608, to M.C.W.), the Bundesministerium für Bildung und Forschung (ICMR2019-016 to M.C.W.), the Berlin University Alliance (501_BIS-CryoFac to M.C.W.), the Indian Council of Medical Research (AMR/INDO/GER/219/2019-ECD-IIG to R.S.), and the National Institutes of Health (GM067153 to I.A.).

## Author contributions

D.G. produced recombinant proteins and assembled complexes with help by N.S. T.H. acquired, processed and refined cryoEM data. D.G. and B.L. built and refined atomic models. D.G. conducted in vitro biochemical and biophysical experiments. N.K. cloned constructs for in vivo studies and conducted the in vivo pull-down, bacterial growth, termination, and β-galactosidase assays. B.W. performed bioinformatic analyses. All authors contributed to the analysis of the data and the interpretation of the results. D.G. wrote the original draft with

contributions from I.A. and R.S. M.C.W. revised the manuscript with contributions from D.G., I.A., and R.S. I.A., R.S., and M.C.W. supervised work in their respective groups and coordinated the collaboration.

## Funding

## Competing interests

The authors declare no competing interests.
