## [Peer Review file · Nature Communications]

The Psu protein of phage satellite P4 inhibits transcription termination factor ρ by forced hyper-oligomerization

Corresponding Author: Professor Markus Wahl

Editorial Note: This manuscript has been previously reviewed at another journal. This document only contains information relating to versions considered at Nature Communications.

Editorial Note: Parts of this Peer Review File have been redacted as indicated to remove third party material where no permission to publish were obtained.

Version 1:

Reviewer comments:

Reviewer #1

(Remarks to the Author)

Gjorgjevikj et al. have addressed my initial concerns. I recommend acceptance once they address the following question and suggestion.

1. One lingering concern is why are the clash scores so high- the pdb validation reports report an error. Please fix.
2. Fig 3 models should have maps accompanying them in transparent surfaces.

Reviewer #2

(Remarks to the Author)

Reviewer #4

(Remarks to the Author)

Review of "Transcription termination factor ρ polymerizes under stress" by Gjorgjevikj et al. 2024.

As I wrote in my initial review, my overall opinion of this manuscript is positive. I think it is suitable for publication in a high quality, broad scope journal. Given my technical expertise, I will limit my review almost entirely to the comparative sequence analysis, which is a rather minor aspect of the manuscript.

In the rebuttal and revised manuscript, the authors have attempted to respond to my comments. I thank them for their efforts.

However, even after reading the rebuttal, I am still not convinced of the relevance of the section "P4-like mobile elements encoding Psu are hubs for phage defense genes". At least as written, I think it has no place in the main text of the manuscript.

First, this section could only be relevant to the manuscript if the authors provide some evidence (possibly from the literature) that Psu regulates the expression of the region between psu and int that contains the phage-defense systems. This is not clear from the current content of the manuscript.

This opinion is somewhat consistent with two of the other three reviewers. In their general comments, reviewers #3 and #5 found it "a bit untested and disconnected from the rest of the manuscript" and "unconvincing" respectively. I also fully agree with reviewer #5's detailed comments on this section. In particular, the text of this section needs to be more clearly positioned in relation to previous work:

* Rousset et al. 2022 have already characterised the region between *psu* and *int* as a reservoir of phage defense systems in *E. coli*, and they have already described the presence of genes encoding sirtuin (Sir2)-HerA adjacent to *psu* in this bacterium.

* de Sousa and Rocha 2022 have already analysed the conservation of P4-like satellite phages on a broader phylogenetic scale.

Second, in my initial review I expressed concerns about the computational workflow for identifying *Psu* homologs, and I do not believe that these have been satisfactorily addressed in the current version of the manuscript. In particular, I remain puzzled by the complexity of the proposed workflow, which the authors justify by arguing that the search of the PFAM model (PF07455) yielded only 28 sequences. However, de Sousa and Rocha (2022) have successfully used this simple approach to identify occurrences of the *psu* gene not only in Enterobacteriaceae but also in some other bacterial families (at least Erwiniaceae and Hafniaceae). These hits outside Enterobacteriaceae were not found in this manuscript, suggesting that the simple approach is no less sensitive than the cumbersome greedy approach proposed here. In my opinion, if this section (not central to the publication) is kept in the manuscript, the approach should either be simplified or described in more detail (both the method and the results, probably not in the main text).

Finally, this is a minor comment, but as I pointed out in my initial review, the use of the words "mutualistic" and "parasitic" in the sentence "... parasitic satellites exist in mutualistic relationships..." sounds self-contradictory. If the relationships are mutualistic, then the satellites are no longer parasitic. I simply suggest deleting the word "parasitic".

Reviewer #5

(Remarks to the Author)

This is my second time reviewing this work by Gjorgjevikj et al. which has now been retitled 'The *Psu* protein of phage satellite P4 inhibits transcription termination factor ρ by forced hyper-oligomerization'. I want to reiterate my enthusiasm for this work describing the mechanism of ρ inhibition by *psu*, a long standing question in bacteriophage biology that involves fundamental gene expression processes in *E. coli*. That being said, I am not an expert in structural biology or biochemistry, so my ability to critically evaluate those aspects of this work is limited. I remain skeptical of a functional relationship between *Psu* and anti-phage defense systems on P4-like elements, but the authors have moderated their language on this point. There is certainly a spatial relationship between *psu* and anti-phage defense systems, and while this relationship has been previously reported in *E. coli*, the authors' analysis have added to our knowledge of this relationship by surveying P4-like elements in other Enterobacteriaceae taxa. In general, I find that the authors have addressed my previous review's comments in a satisfactory way.

Something I have realized since my initial review is that P4's terminal *cos* site is located between *psu* and P4's defense genes. While the defense genes and *psu* are co-localized in an integrated and circular P4 genome, the linearized P4 episome will have these genes encoded on the opposite ends of the genome. I suspect this is the actual reason for *psu*'s proximity to the defense system genes. The linear P4 genome terminates with a structural operon where *psu* is the final gene. On the other end of the *cos* site/genome the defense system genes are located. This organization is logical because terminal regions are often prone to genetic recombination, and diversification or turnover of this anti-phage locus is likely selected for. My hypothesis is speculative, so I'm not asking that the authors propose it in their manuscript, but I think an analysis of the spatial relationship of *psu* and the defense genes should mention the location of the *cos* site as well. In Figure 9, it would be beneficial to label the *cos* site in P4, as well as in the other genomes where it can be detected.

I have outlined additional minor comments below:

1. In line 32, the authors wrote "Psu may promote expression of anti-phage genes". It is still unintuitive to me why *Psu* would promote anti-phage gene expression when it is expressed during phage replication. I suggest changing 'promote' to a more ambiguous 'regulate'. Lines 39 through 41 mention a number of processes inhibited by ρ that can themselves be inhibitory towards gene transcription, so it's not inconceivable that *psu* could downregulate certain genes through the inhibition of ρ -dependent termination.
2. In line 68, the authors wrote "Phage satellites are mobile genetic elements (MGEs) that hijack structural proteins of helper phages to assemble viral capsids of reduced size." This is technically an overgeneralization. Phage satellites are virus-like entities that require infection by another phage to reproduce. The vast majority of described satellite systems do assemble capsids of reduced size, but there are some notable exceptions that do not remodel capsids or encode their own capsids. Please correct the wording in this sentence. Otherwise this overview of P4 biology is good.
3. In Figure 6 panel A., there is a large number of genomes where only *psu* is detected. The authors should check that these aren't linearized genomes terminating with *psu*, since that could explain those results.
4. As mentioned above, please label detectable *cos* sites in Figure 9 C.

5. Also in Figure 9 C, the authors have added in percent identity values for sid and delta. The legend does not say if these are amino acid or nucleotide identity (I assume amino acid). The delta gene from P4 is notably larger than its homologs in other P4-like elements. Can the authors specify what the homologous region of delta is in the figure? Is it one of the termini?

Sincerely,
Zach Barth

Reviewer #6

(Remarks to the Author)

The authors should be commended on generating a comprehensive, complete, and clear revision of the manuscript in response to reviewers suggestions. There are no remaining concerns with the work and the manuscript seems ready for publication once the authors have a chance to incorporate any final revisions they might want to make.

Version 2:

Reviewer comments:

Reviewer #4

(Remarks to the Author)

The authors have satisfactorily addressed my comments, expected for the Method subsection "Compiling Psu sequences" (L768), for which I wrote in my previous comment "if this section (not central to the publication) is kept in the manuscript, the approach should either be simplified or described in more detail (both the method and the results, probably not in the main text)."

So I will try to comment on this subsection in more detail below.

L769-770 As explained in the rebuttal the authors removed one sentence on their "Reference Proteomes search". It was replaced by a new sentence "A phylogenetic study of P4-like bacteriophage revealed that Psu homologs are limited (~80 % of the analyzed Psu sequences) to three genera: Escherichia, Salmonella, and Klebsiella.". I do not see the interest of this statement in the method section and the need for a "greedy search" remains unclear to me (L771), but a more detailed presentation of the results might justify the interest of the proposed approach.

L771-772 "The E-value cutoffs were estimated based on database size throughout our analysis." What does this mean ? Why the word cutoff ? A cutoff is not "estimated". In contrast, the estimation of an "E-value"ⁿ by definition, takes into account the size of the database. It is not clear whether the sentence is informative ?

L771-773 "An E-value of 10⁻⁴ was used as a starting point as suggested in reference 98." It seems that the authors refer here to the "cutoff" value applied to E-values to select hits. Clarify. Also why do they mention "a starting point" ? It seems that this is the only cutoff on the E-value used in analysis ? also please format properly the reference.

L774-775 "to fish out all Psu candidates". The verb "fish out" does not seem appropriate in academic writing. Nor do I see the need to write "all".

L780 "Jackhmmer will build" why the future tense is used here ?

L781-782 "After seven iterations, the number of gathered sequences did not change anymore; this step yielded 2556 sequences." It is unclear what the overlap is between the 2556 sequences mentioned here, retrieved with Jackhmmer, and the 908 sequences mentioned in L778, retrieved with hmmsearch. Clarify. The use of the term "step" for the Jackhmmer search is also inconsistent with its use in L773 ("The "greedy search" is composed of two steps").

L782-785. "To find potentially missed homologs by the profile-based searching, we deployed blastp searching with Psu reference sequence as a query against UniProtKB (v.2022_04) on the UniProt website (<https://www.uniprot.org/>) with E-value 10⁻⁴; this step gathered 250 sequences." Again, it is unclear how the 250 overlap with the 908 and 2556 sequences, retrieved with hmmsearch and Jackhmmer, respectively. Also the authors use a different version of UniProtKB here (v.2022_04 vs. v.2021_04). So, it is unclear whether the additional hits are due to the increase in size of the UniProtKB or to the use of a non-"profile-based" algorithm. Clarify. Finally, the verb "deploy" sounds awkward in this context.

L786 "Next, we performed a blastp search of Psu reference sequence against Reference proteins on NCBI (<https://www.ncbi.nlm.nih.gov/>) with E-value 10⁻⁴. Max target sequences option was set to 5000; 2820 sequences were found by this step." How many of these sequences overlap with those found in UniProtKB ? Why did the authors need to search both UniProtKB and NCBI protein databases ? If there is an interest in also searching the NCBI "Reference proteins" why did they not use the in principle more sensitive "profile-based" approaches used to find hits in UniProtKB ? It is also unclear whether more hits could have been found by increasing the value of the "Max target sequences" option ? What are "Reference proteins on NCBI" database mentioned here ? Are they the "RefSeq non-redundant proteins"

(<https://www.ncbi.nlm.nih.gov/refseq/about/nonredundantproteins/>) or the "Protein database" (<https://www.ncbi.nlm.nih.gov/protein/>) ?

L788-791 "As Psu sequences from the phage were scarce, we attempted to find more viral Psu proteins using a search with Psu reference sequence as a query against Non-redundant protein database on NCBI with E-value 10⁻⁴. The search was limited to the Virus superkingdom (Taxonomy ID: 10239). " Why did the authors expect to identify more hits by using of this "non-redundant protein database" (see also the question above about the exact database mentioned L786)? Clarify if it is because of its smaller size (due to its restricted taxonomic range or non-redundancy" which induces a lower E-value for the same hit ? In this case, this does not seem to be the most desirable.

L790-792 "Max target sequences option was set to 5000. Only 4 sequences were found in this step." What is the effect of the value "Max target sequences option"? With only 4 (additional?) hits identified, this last line of search does not seem relevant for the main text.

Reviewer #5

(Remarks to the Author)

I have looked over the changes in the latest version of the manuscript. There is a small discrepancy between the submitted file and the response to my comments. In their response, the authors write

"We rephrased this statement to read (line 68):

Phage satellites are virus-like entities that require infection by another (helper) phage to reproduce. The majority of phage satellites, including P4, hijack structural proteins of helper phages to assemble viral capsids of reduced size."

However, the sentence in the submitted manuscript is missing the clause "hijack structural proteins of helper phages to". I think this clause contains important information and should be included in the final text. Since it was included in the author response, I assume it was omitted in the submitted text out of error.

Other than this error, all of my comments have been satisfactorily addressed.

Response to Reviewer Comments

Reviewer comments – bold

Our responses – red

Changed text passages – highlighted in yellow

Coordinate and map files as well as PDB validation reports are available at:

[Redacted]

Reviewer #1

Gjorgjevikj et al. have addressed my initial concerns. I recommend acceptance once they address the following question and suggestion.

We thank the reviewer for this assessment.

1. One lingering concern is why are the clash scores so high - the pdb validation reports report an error. Please fix.

All PDB validation reports we received upon submitting our models to the PDB, except the report for the p^{P167L}-ATPyS structure, were returned without the clashscore analysis. Instead they state: “Due to software issues we are unable to calculate clashes - this section is therefore empty.” However, this is not due to an excessive number of close contacts in our models, but rather seems to be a computational issue on the side of the PDB, possibly due to the very large number of atoms contained in the models. We calculated clashscores with Molprobity. Clashscores for our structural models fall between 12.5 and 18.8, as reported in Supplementary Table 1. Below, we provide a figure and a table from PMID 22000512, documenting that these values fall well within the “Good Quartile” of structural models in the PDB, and that the median clashscore for PDB structures at 3 Å resolution is 39, well above the values for our models. Reasonable clashscores for our structures are also indicated by the comparatively small number of close contacts that are listed in the headers of the PDB files.

[Redacted]

Therefore, there is no problem with high clashscores in our models and, thus, we see nothing we could fix regarding this issue.

2. Fig 3 models should have maps accompanying them in transparent surfaces.

We experimented extensively with showing the map as a transparent surface or as a mesh. We were not satisfied with either version, for mesh here is an example:

The reviewers had inspected the maps. We also have tested roles of individual residues by site-directed mutagenesis and functional assays, and obtained results fully consistent with the structural models. For sake of clarity, we would therefore prefer to keep Fig. 3 models as they are.

Reviewer #2

The reviewer did not raise further issues.

Reviewer #4

Review of "Transcription termination factor ρ polymerizes under stress" by Gjorgjevikj et al. 2024.

As I wrote in my initial review, my overall opinion of this manuscript is positive. I think it is suitable for publication in a high quality, broad scope journal. Given my technical expertise, I will limit my review almost entirely to the comparative sequence analysis, which is a rather minor aspect of the manuscript.

We thank the reviewer for the overall positive evaluation.

In the rebuttal and revised manuscript, the authors have attempted to respond to my comments. I thank them for their efforts.

However, even after reading the rebuttal, I am still not convinced of the relevance of the section "P4-like mobile elements encoding *Psu* are hubs for phage defense genes". At least as written, I think it has no place in the main text of the manuscript.

The section "P4-like mobile elements encoding Psu are hubs for phage defense genes" has been moved to the Supplementary Discussion.

First, this section could only be relevant to the manuscript if the authors provide some evidence (possibly from the literature) that Psu regulates the expression of the region between *psu* and *int* that contains the phage-defense systems. This is not clear from the current content of the manuscript.

The molecular mechanism of Psu is distinct from those of canonical phage anti-terminators, such as phage λ N and O proteins, which are recruited to transcribing RNAP *in cis*, through interactions with specific RNA or DNA elements, and activate expression of genes downstream of these signals. Psu binds to p and can thus act on any gene that is susceptible to p-mediated termination. E.g., Psu acts *in trans* to rescue amber mutants of P2 (PMID 789895) and to inhibit termination at diverse p-dependent signals placed on the chromosome or plasmids (PMIDs 3540944, 9007066, and many others). Furthermore, we and others use Psu (and its E56K variant as a control) in place of bicyclomycin, a specific inhibitor of p that is prohibitively expensive, to inhibit p *in vivo*.

It is possible that Psu acts more efficiently when produced *in cis*. Psu activates expression of P4 late genes *in cis*, as described by Lagos *et al.* (PMID 3540944); the burst of P4 *psu* mutants is reduced three- to fivefold relative to that of P4 *psu+* (PMID 7021852). Therefore, we expect that Psu would promote expression of phage-defense cassettes that are inserted downstream of the late genes, extending the late operon.

p is known to silence horizontally-acquired genes, which tend to be more AT-rich than the surrounding resident DNA. Notably, the defense cassettes identified here are strikingly more AT-rich than their linked P4-like elements, with differences ranging from 15 to 25 %. Therefore, by analogy to other known p targets (prophages, EPODs, etc.), we hypothesize that p silences these genes while Psu counter-silences p. We clarified this point in the Supplementary Discussion section.

To provide evidence from the literature that Psu regulates expression of the region between *psu* and *int* genes, we included a brief summary of the above in the Discussion section of the revised manuscript (line 509):

Many defense genes are codirectional with *psu* (Fig. 9), potentially extending the *psu* operon, which is known to be activated by Psu³⁰. Notably, the molecular mechanism of Psu is distinct from those of canonical phage anti-terminators, such as phage λ N and O proteins, which are recruited to transcribing RNAP at specific RNA or DNA elements and activate expression of genes downstream of these elements⁸². Psu binds to p and can thus act on any gene that is susceptible to p-mediated termination, possibly more efficiently when produced *in cis*.

This opinion is somewhat consistent with two of the other three reviewers. In their general comments, reviewers #3 and #5 found it "a bit untested and disconnected from the rest of the manuscript" and "unconvincing" respectively. I also fully agree with reviewer #5's detailed comments on this section. In particular, the text of this section needs to be more clearly positioned in relation to previous work:

* Rousset *et al.* 2022 have already characterised the region between *psu* and *int* as a reservoir of phage defense systems in *E. coli*, and they have already described the presence of genes encoding sirtuin (Sir2)-HerA adjacent to *psu* in this bacterium.

* de Sousa and Rocha 2022 have already analysed the conservation of P4-like satellite phages on a broader phylogenetic scale.

We elucidated a mechanism whereby Psu modulates activity of ρ . The mode of Psu action is novel and could be used by other regulators that promote ρ oligomerization. However, we think that pointing out that Psu is associated with known or putative defense systems in three quarters of the genomes it is found in would be of interest to a much broader audience. In our work, we expanded the findings beyond *E. coli* and its close relatives and were able to identify numerous putative defense systems that DefenseFinder misses. Given the diversity of defense systems, another means of context-dependent identification, as provided here, would be of value.

To address the reviewer's concerns, we moved the relevant sections to the Supplementary Information, kept only a short description of our findings in the Discussion section of the main text and more clearly positioned this description in relation to previous work by de Sousa *et al.* and Rousset *et al.* (references 76 and 77 of the revised manuscript; line 497):

P4-like elements found in enterobacterial genomes frequently carry phage defense systems^{76,77}, some of which can protect *E. coli* from lytic phages, including λ , P1, and T7, while not restricting P2⁷⁷. P4 can infect diverse bacteria, prompting us to search for Psu homologs across bacteria (see Supplementary Discussion). Our analysis revealed that more than three quarters of *psu*^r genomes, including those outside of *Escherichia*, contain either a known⁷⁸ or a putative defense system inserted between the *psu* and *int* genes of P4 (Supplementary Fig. 13).

Second, in my initial review I expressed concerns about the computational workflow for identifying Psu homologs, and I do not believe that these have been satisfactorily addressed in the current version of the manuscript. In particular, I remain puzzled by the complexity of the proposed workflow, which the authors justify by arguing that the search of the PFAM model (PF07455) yielded only 28 sequences. However, de Sousa and Rocha (2022) have successfully used this simple approach to identify occurrences of the *psu* gene not only in Enterobacteriaceae but also in some other bacterial families (at least Erwiniaceae and Hafniaceae). These hits outside Enterobacteriaceae were not found in this manuscript, suggesting that the simple approach is no less sensitive than the cumbersome greedy approach proposed here. In my opinion, if this section (not central to the publication) is kept in the manuscript, the approach should either be simplified or described in more detail (both the method and the results, probably not in the main text).

Association of RM defense systems with P4-like elements has been known since the late 1980s (PMIDs 2836359, 9628335, 10542186 etc.). However, since P4/P2 can infect bacteria outside of *Escherichia*, we wanted to expand the searches by de Sousa, Rocha, Rousset and coworkers. We are not arguing that the Psu model (PF07455) cannot find homologs. Our initial search with the Psu model (PF07455) was done in Reference Proteomes (v.2021_04), which is why only 28 sequences were returned. However, an unavoidable shortcoming for all models, this Psu model might fail to identify distant homologs. The limited number and taxonomy distribution of Psu hits motivated us to use a more comprehensive search combination. We removed the description of Reference Proteomes search from the Methods and added (line 769):

A phylogenetic study of P4-like bacteriophage revealed that Psu homologs are limited (~80 % of the analyzed Psu sequences) to three genera: *Escherichia*, *Salmonella*, and *Klebsiella*.⁷⁶ To obtain a comprehensive picture of Psu diversity, we launched a "greedy search".

Regarding host range, we used the Genome Taxonomy Database (GTDB; PMID: 30148503; PMID: 34520557) taxonomy, while de Sousa and Rocha (2022) used NCBI taxonomy. The GTDB is an initiative to establish a standardized microbial taxonomy based on genome phylogeny. All the families outside Enterobacteriaceae discussed in de Sousa and Rocha (2022) are absorbed into Enterobacteriaceae in the GTDB taxonomy. For example: *Pantoea vagans*

(GCF_000148935.1) from Erwiniaceae, *Edwardsiella* sp. (GCA_000800725.2) from Hafniaceae, *Pectobacterium wasabiae* (GCF_001742185.1) from Pectobacteriaceae, and *Serratia* sp. (GCF_003691565.1) from Yersiniaceae are assigned to Enterobacteriaceae in GTDB. So, de Sousa and Rocha (2022) did not find *Psu* outside Enterobacteriaceae.

Finally, using only PF07455, de Sousa and Rocha (2022) identified ~1000 *Psu* sequences, whereas our search identified ~3000 *Psu* sequences. Most (~84 %) of the *Psu* sequences found by de Sousa and Rocha (2022) come from genomes belonging to genera *Escherichia*, *Salmonella* and *Klebsiella*. In our results, ~60 % of *Psu* sequences belong to these three genera. Most importantly, taking advantage of the newly mined *psu* genes, we identified numerous known defense systems as well as many potential novel systems.

For the reasons provided above, we believe that our approach and findings are valuable extensions of the previous work. We think that our approach is appropriately described in the manuscript, and therefore retained its description in the revised manuscript, while moving the relevant results section to the Supplementary Information. We revised the main text Discussion to elaborate our hypothesis on *Psu*-mediated regulation of associated defense systems. We would prefer to retain an illustration of the P4-associated defense cassettes (original Fig. 9c), including a putative system identified in this work, with modifications suggested by Reviewer #5 – we think that these results are of interest to a broad audience of *Nature Communications*.

Finally, this is a minor comment, but as I pointed out in my initial review, the use of the words "mutualistic" and "parasitic" in the sentence "... parasitic satellites exist in mutualistic relationships..." sounds self-contradictory. If the relationships are mutualistic, then the satellites are no longer parasitic. I simply suggest deleting the word "parasitic".

This sentence was deleted from the revised manuscript.

Reviewer #5

This is my second time reviewing this work by Gjorgjevikj et al. which has now been retitled 'The *Psu* protein of phage satellite P4 inhibits transcription termination factor ρ by forced hyper-oligomerization'. I want to reiterate my enthusiasm for this work describing the mechanism of ρ inhibition by *psu*, a long standing question in bacteriophage biology that involves fundamental gene expression processes in *E. coli*. That being said, I am not an expert in structural biology or biochemistry, so my ability to critically evaluate those aspects of this work is limited. I remain skeptical of a functional relationship between *Psu* and anti-phage defense systems on P4-like elements, but the authors have moderated their language on this point. There is certainly a spatial relationship between *psu* and anti-phage defense systems, and while this relationship has been previously reported in *E. coli*, the authors' analysis have added to our knowledge of this relationship by surveying P4-like elements in other Enterobacteriaceae taxa. In general, I find that the authors have addressed my previous review's comments in a satisfactory way.

We thank the reviewer for the very positive assessment and for considering their original comments well-addressed.

Something I have realized since my initial review is that P4's terminal *cos* site is located between *psu* and P4's defense genes. While the defense genes and *psu* are co-localized in an integrated and circular P4 genome, the linearized P4 episome will have these genes encoded on the opposite ends of the genome. I suspect this is the actual reason for *psu*'s proximity to the defense system genes. The linear P4 genome terminates with a structural operon where *psu* is the final

gene. On the other end of the *cos* site/genome the defense system genes are located. This organization is logical because terminal regions are often prone to genetic recombination, and diversification or turnover of this anti-phage locus is likely selected for. My hypothesis is speculative, so I'm not asking that the authors propose it in their manuscript, but I think an analysis of the spatial relationship of *psu* and the defense genes should mention the location of the *cos* site as well. In Figure 9, it would be beneficial to label the *cos* site in P4, as well as in the other genomes where it can be detected.

We thank the reviewer for this idea. The *cos* sites have been detected in all 6 representatives and are indicated in the revised Fig. 9, as suggested. Below is the revised Fig. 9 (please note that former Fig. 9 panels a and b have been moved to the revised Supplementary Fig. 13) and the relevant part of the legend (line 1303):

... The locations of *cos* sites (black boxes) were identified by multiple sequence alignments of DNA sequences within 300 bp downstream of the *psu* gene using Clustal Omega¹⁰⁸ and *cos* sequences from reference¹⁰⁹. ...

I have outlined additional minor comments below:

1. In line 32, the authors wrote “*Psu* may promote expression of anti-phage genes”. It is still unintuitive to me why *Psu* would promote anti-phage gene expression when it is expressed during phage replication. I suggest changing ‘promote’ to a more ambiguous ‘regulate’. Lines 39 through 41 mention a number of processes inhibited by *rho* that can themselves be inhibitory towards gene transcription, so it’s not inconceivable that *psu* could downregulate certain genes through the inhibition of *rho*-dependent termination.

As also stated in response to Reviewer #2, our reasoning was that *Psu* is likely to act *in cis*, as it does in P4, thereby directly inhibiting *p* and activating expression of cassettes it is linked to. The defense genes identified here are very AT-rich (15-25 % difference from the *sid-delta-psu* operon) and are likely targets for *p*, which has been shown to silence AT-rich regions in *E. coli* and *Salmonella*. We agree that, given many effects of *p* genome wide, *Psu* may also have an opposite effect in some cases, and changed “promote” to neutral “regulate” (line 32).

2. In line 68, the authors wrote “Phage satellites are mobile genetic elements (MGEs) that hijack structural proteins of helper phages to assemble viral capsids of reduced size.” This is technically

an overgeneralization. Phage satellites are virus-like entities that require infection by another phage to reproduce. The vast majority of described satellite systems do assemble capsids of reduced size, but there are some notable exceptions that do not remodel capsids or encode their own capsids. Please correct the wording in this sentence. Otherwise this overview of P4 biology is good.

We rephrased this statement to read (line 68):

Phage satellites are virus-like entities that require infection by another (helper) phage to reproduce. The majority of phage satellites, including P4, hijack structural proteins of helper phages to assemble viral capsids of reduced size.

3. In Figure 6 panel A, there is a large number of genomes where only *psu* is detected. The authors should check that these aren't linearized genomes terminating with *psu*, since that could explain those results.

Yes, this is correct. -60 % of the lonely *psu* genes are at the genome terminus. We revised the legend of Supplementary Fig. 13 to clarify the point (former Fig. 9a was moved to the revised Supplementary Fig. 13, panel b; SI, line 308):

We note that the absence of the *int* gene could be due to incomplete genome assembly in some cases and -60 % *psu* genes in the "*psu* alone" group are located at the terminus of linearized genomes.

4. As mentioned above, please label detectable *cos* sites in Figure 9 C. Done;
for revised Fig. 9 and the relevant part of the legend please see above.

5. Also in Figure 9 C, the authors have added in percent identity values for *sid* and *delta*. The legend does not say if these are amino acid or nucleotide identity (I assume amino acid). The *delta* gene from P4 is notably larger than its homologs in other P4-like elements. Can the authors specify what the homologous region of *delta* is in the figure? Is it one of the termini?

They are amino acid residue identities, which is now explained in the revised Fig. 9 legend (line 1300):

Numbers indicate percent amino acid residue identities to P4 *sid*, *delta*, *psu*, and *int*-encoded proteins.

The newly pulled *Delta* homologs cover -50 % of the P4 *Delta*. The homologous regions include residues 1-40 and 120-160 of P4 *Delta* (NP_042043.1). Residues 1-40 harbor the recombinase zinc beta ribbon domain of P4 *Delta*. The new homologs cover both N- and C-terminus of P4 *Delta*. We now specify this in the revised Fig. 9 legend (line 1301):

Delta homologs shorter than the P4 *Delta* protein (NP_042043.1) contain homologous regions corresponding to residues 1-40 and 120-160 of P4 *Delta*.

Sincerely,
Zach Barth

Reviewer #6

The authors should be commended on generating a comprehensive, complete, and clear revision of the manuscript in response to reviewers suggestions. There are no remaining concerns with the work and the manuscript seems ready for publication once the authors have a chance to incorporate any final revisions they might want to make.

We thank the reviewer for considering our revision comprehensive, complete and clear.

Response to Reviewer Comments

Reviewer comments – bold

Our responses – red

Changed text passages – highlighted in yellow

Coordinate and map files as well as PDB validation reports are available at:

[Redacted]

Reviewer #4

The authors have satisfactorily addressed my comments, expected for the Method subsection “Compiling Psu sequences” (L768), for which I wrote in my previous comment “if this section (not central to the publication) is kept in the manuscript, the approach should either be simplified or described in more detail (both the method and the results, probably not in the main text).”

So I will try to comment on this subsection in more detail below.

L769-770 As explained in the rebuttal the authors removed one sentence on their “Reference Proteomes search”. It was replaced by a new sentence “A phylogenetic study of P4-like bacteriophage revealed that Psu homologs are limited (~80 % of the analyzed Psu sequences) to three genera: Escherichia, Salmonella, and Klebsiella.”. I do not see the interest of this statement in the method section and the need for a “greedy search” remains unclear to me (L771), but a more detailed presentation of the results might justify the interest of the proposed approach.

We clearly failed to interpret the reviewer’s comment correctly. In the revised manuscript, we provided justification for launching a wide-net search for Psu homologs, simplified the description, and provided responses to the reviewer’s questions. Additional details that were introduced in response to the first round of reviewers’ comments were either deleted or streamlined.

In our study, we have cast a wide net to identify psu-like genes. As a result, we have identified not only known defense systems but also many potential novel systems. We think that this information is of interest to a broad audience of Nature Communications.

The term “greedy search” was removed to avoid confusion. In this study, we did not design any special searching method. We searched the databases using the online tools on the websites mentioned in the Methods.

L771-772 “The E-value cutoffs were estimated based on database size throughout our analysis.” What does this mean? Why the word cutoff? A cutoff is not “estimated”. In contrast, the estimation of an “E-value”, by definition, takes into account the size of the database. It is not clear whether the sentence is informative?

In the first round of review, we were asked for reasons behind E-value selection and tried to explain them. We removed this sentence and simplified our description to state (line 783):

An E-value of 10^{-4} was used during searches of large databases.

L771-773 “An E-value of 10⁻⁴ was used as a starting point as suggested in reference 98.” It seems that the authors refer here to the “cutoff” value applied to E-values to select hits. Clarify. Also why do they mention “a starting point”? It seems that this is the only cutoff on the E-value used in analysis? Also please format properly the reference.

“An E-value of 10⁻⁴ was used as a starting point as suggested in reference 98.” was deleted. We explicitly stated E-values used for large and small datasets in the revised manuscript.

L774-775 “to fish out all Psu candidates”. The verb “fish out” does not seem appropriate in academic writing. Nor do I see the need to write “all”.

Changed to (line 771):

To explore this possibility, we sought to identify as many Psu homologs as possible in two steps: ...

L780-782 “Jackhmmer will build” why the future tense is used here? “After seven iterations, the number of gathered sequences did not change anymore; this step yielded 2556 sequences.” It is unclear what the overlap is between the 2556 sequences mentioned here, retrieved with Jackhmmer, and the 908 sequences mentioned in L778, retrieved with hmmsearch. Clarify. The use of the term “step” for the Jackhmmer search is also inconsistent with its use in L773 (“The “greedy search” is composed of two steps”).

Changed to (line 776):

Jackhmmer attempts to find more homologs by building a new HMM at each algorithm iteration; in our study, the total number of retrieved sequences did not change after seven iterations, suggesting convergence.

In our workflow, all sequences were pooled before downloading, and identical sequences were removed. Indeed, the hmmsearch identified a subset of 2556 jackhmmer hits. We deleted the hmmsearch from this subsection.

L782-785. “To find potentially missed homologs by the profile-based searching, we deployed blastp searching with Psu reference sequence as a query against UniProtKB (v.2022_04) on the UniProt website (<https://www.uniprot.org/>) with E-value 10⁻⁴; this step gathered 250 sequences.” Again, it is unclear how the 250 overlap with the 908 and 2556 sequences, retrieved with hmmsearch and Jackhmmer, respectively. Also the authors use a different version of UniProtKB here (v.2022_04 vs. v.2021_04). So, it is unclear whether the additional hits are due to the increase in size of the UniProtKB or to the use of a non-“profile-based” algorithm. Clarify. Finally, the verb “deploy” sounds awkward in this context.

About 80 % of the 250 sequences from the UniProt website overlapped with the jackhmmer result. Since only 7 out of the 45 additional hits from UniProt were due to the non-profile-based method, we deleted the blastp search against UniProtKB to simplify the description.

The jackhmmer was done on HmmerWeb (v. 2.41.2), which used UniProtKB (v.2021_04). And subsequently we performed a blastp search in the UniProt website, which had the latest version (v.2022_04) of the database at the time of searching.

L786 “Next, we performed a blastp search of Psu reference sequence against Reference proteins on NCBI (<https://www.ncbi.nlm.nih.gov/>) with E-value 10⁻⁴. Max target sequences option was set to 5000; 2820 sequences were found by this step.” How many of these sequences overlap with those found in UniProtKB ? Why did the authors need to search both UniProtKB and NCBI protein databases ? If there is an interest in also searching the NCBI “Reference proteins” why did they not use the in principle more sensitive “profile-based” approaches used to find hits in UniProtKB ? It is also unclear whether more hits could have been found by increasing the value of the “Max target sequences” option ? What are “Reference proteins on NCBI” database mentioned here ? Are they the “RefSeq non-redundant proteins” (<https://www.ncbi.nlm.nih.gov/refseq/about/nonredundantproteins/>) or the “Protein database” (<https://www.ncbi.nlm.nih.gov/protein/>)?

288 of the 2820 sequences collected from NCBI are identical to those from UniProtKB. We removed all intermediate numbers from the revised manuscript and summarized the results after Psu homolog verification (line 790):

Finally, 2,624 Psu sequences were kept (Supplementary Dataset 1A). Among them, 2,523 of Psu homologs were obtained from the NCBI database and an additional 101 sequences were retrieved from the UniProtKB database.

UniProtKB and NCBI databases have different sample sizes and also produce different hits. As of Nov. 14th 2024, UniProtKB contains ~ 249 million sequences, while NCBI RefSeq_protein database contains ~365 million sequences. We used both databases to identify as many hits as possible, as stated in the revised manuscript.

Limited by our computing resources, we did the blastp search on the NCBI website where the profile-based method is not available. Profile-based search was done on HmmerWeb server, which uses UniProtKB.

Increasing the value of “Max target sequences” will not yield more hits and was deleted.

Yes, they are the “RefSeq non-redundant proteins”. RefSeq non-redundant protein database is now used in this subsection.

L788-791 “As Psu sequences from the phage were scarce, we attempted to find more viral Psu proteins using a search with Psu reference sequence as a query against Non-redundant protein database on NCBI with E-value 10⁻⁴. The search was limited to the Virus superkingdom (Taxonomy ID: 10239). ” Why did the authors expect to identify more hits by using of this “non-redundant protein database” (see also the question above about the exact database mentioned L786)? Clarify if it is because of its smaller size (due to its restricted taxonomic range or non-redundancy” which induces a lower E-value for the same hit? In this case, this does not seem to be the most desirable.

The non-redundant protein database is a superset of RefSeq non-redundant proteins database (~840 million vs 365 million as of Nov. 14th 2024). This expanded database contains all non-redundant GenBank CDS translations + PDB (Protein Data Bank) + SwissProt + PIR (Protein Information Resource) + PRF (Protein Research Foundation) excluding environmental samples from WGS projects.

L790-792 “Max target sequences option was set to 5000. Only 4 sequences were found in this step.” What is the effect of the value “Max target sequences option”? With only 4 (additional?) hits identified, this last line of search does not seem relevant for the main text.

“Max target sequences option was set to 5000” was deleted. We think that the fact that we identified only 4 additional sequences indicates that we have found nearly all Psu homologs in currently available databases.

Reviewer #5

I have looked over the changes in the latest version of the manuscript. There is a small discrepancy between the submitted file and the response to my comments. In their response, the authors write

"We rephrased this statement to read (line 68):

Phage satellites are virus-like entities that require infection by another (helper) phage to reproduce. The majority of phage satellites, including P4, hijack structural proteins of helper phages to assemble viral capsids of reduced size."

However, the sentence in the submitted manuscript is missing the clause "hijack structural proteins of helper phages to". I think this clause contains important information and should be included in the final text. Since it was included in the author response, I assume it was omitted in the submitted text out of error.

Other than this error, all of my comments have been satisfactorily addressed.

Sorry for this mishap, we added the clause “hijack structural proteins of helper phages to” (line 69).